# Staphylococcal Enterotoxin M Exhibits Thrombin-like Enzymatic Activity

**DOI:** 10.3390/biom15101357

**Published:** 2025-09-24

**Authors:** Qian Huang, Shuang-Hua Luo, Wan-Fan Tian, Jun-Ni Tang, Ji Liu

**Affiliations:** 1College of Pharmacy and Food, Southwest Minzu University, Chengdu 610041, China; huangqian0904@gmail.com (Q.H.); shluo2017@swun.edn.cn (S.-H.L.); tianwanfan2016@swun.edu.cn (W.-F.T.); 2College of Animal and Veterinary Sciences, Southwest Minzu University, Chengdu 610041, China

**Keywords:** staphylococcal enterotoxin M, His_6×_-TCS-ΔNspSEM_WT_ fusion protein, thrombin-like activity, thrombin cleavage site sequence (LVPR↓GS), immobilized metal affinity chromatography, chromogenic substrate S-2238, size-exclusion high-performance liquid chromatography, AlphaFold 3, serine protease triad (H_172_/S_178_/D_212_), mutant His_6×_-TCS-ΔNspSEM_S178A_ fusion protein, molecular dynamics simulations

## Abstract

To express and purify staphylococcal enterotoxin M (SEM) using immobilized metal affinity chromatography (IMAC), a signal peptide-truncated (ΔNsp) wild-type SEM (SEM_WT_) was N-terminally fused in pET-28a(+) to a polyhistidine tag (His_6×_-) and thrombin cleavage site (TCS; LVPR↓GS), generating His_6×_-TCS-ΔNspSEM_WT_. Unexpectedly, 4 °C desalting reduced the fusion protein’s molecular weight by ~2.0 kDa on sodium dodecyl sulfate-polyacrylamide gel electrophoresis (SDS-PAGE). N-terminal sequencing and mass spectrometry identified cleavage specifically at the arginine (R) and glycine (G) peptide bond (R–G bond) within the TCS motif. AlphaFold 3 revealed an exposed serine protease catalytic triad: histidine 172, serine 178, and aspartic acid 212 (H_172_/S_178_/D_212_) in the *β*-grasp domain, suggesting intrinsic thrombin-like activity (TLA). Sequential IMAC and size-exclusion high-performance liquid chromatography (SE-HPLC) purification eliminated contaminant concerns, while chromogenic substrate S-2238 (S-2238) assays demonstrated increasing specific activity and purification fold, supporting intrinsic TLA. Critically, the mutation of serine at position 178 to alanine (His_6×_-TCS-ΔNspSEM_S178A_) abolished TLA but preserved the secondary/tertiary structure, confirming the activity’s origin within the wild-type construct. Molecular dynamics (MD) simulations probed the atomistic mechanism for specific R–G bond cleavage. This work establishes a foundation for understanding ΔNspSEM_WT_’s TLA.

## 1. Introduction

*Staphylococcus aureus* (*S. aureus*), a Gram-positive opportunistic pathogen [1], colonizes respiratory epithelia and skin. Its virulence and antibiotic resistance stem largely from mobile genetic elements (MGEs) encoding critical host adaptation factors [2]. Among diverse virulence determinants, staphylococcal enterotoxins (SEs) represent notable pyrogenic toxin superantigens (SAgs) [3]. With high sequence homology, SEs induce immune hyperactivation, potentially causing toxic shock and death [4]. This SE superfamily comprises 29 members, classified as classical SE serotypes (A, Bn, Cn, D, E, G), staphylococcal enterotoxin-like (SE*l*) variants (H, I, J-X), and toxic shock syndrome toxin-1 (TSST-1). According to International Nomenclature Committee for Staphylococcal Superantigens (INCSS) recommendations [5], the SE/SE*l* nomenclature distinguishes members with demonstrated emetic activity [6].

The enterotoxin gene cluster (*egc*), harbored within genomic island υSa and encoding SEM, was identified in 2001 [7]. Recombinant SEM (rSEM) induces selective T-cell expansion via specific TCR V*β* interactions. Following *egc*’s recognition as a prevalent enterotoxin reservoir, research has focused on SEM-associated pathogenesis [8,9,10,11,12,13], confirming potent T-cell activation [14]. Nevertheless, the structural basis for SEM-mediated TCR V*β*/MHC II binding remains undefined, necessitating future high-resolution ternary complex analysis. Unlike classical SEs, SEM (SEI-group) exhibits weaker emetic potency than SEA [15,16] but contributes to staphylococcal food poisoning (SFP). While SEA’s emetic mechanism involves mucosal translocation, mast cell 5-HT release, and vagal activation [17], SEM’s pathway—particularly mucosal transport and mast cell interactions—remains elusive. In contrast to SEA, SEM’s intestinal translocation remains uncharacterized, although it induces inflammation in bovine mammary epithelia [18].

All SEs share a conserved fold: (1) an N-terminal *β*-barrel domain, (2) a C-terminal *β*-grasp fold, and (3) a central diagonal *α*-helix. Structural variations classify SEs into five evolutionary groups (I–V) [19], with SEM in group V. Group V SAgs (e.g., SEK, SEM, SEQ, SEV [20]) lack the emetic cysteine loop but retain group III/IV-like MHC II binding sites. Crucially, SEM possesses a distinctive 15-residue *α*3-*β*8 loop dictating TCR V*β* specificity and superantigenicity [21]. We encountered constraints studying SEs due to low native protein yield and purity from *S. aureus*. Therefore, we employed recombinant expression in *E. coli*. Following purification by IMAC and desalting via centrifugal ultrafiltration (Section 2.2 and Section 2.3), the engineered His_6×_-TCS-ΔNspSEMWT fusion protein underwent unexpected autocleavage at the TCS R–G bond (LVPR↓GS). This self-proteolytic activity indicates intrinsic TLA within the fusion protein, serendipitously revealing previously unrecognized enzymatic properties meriting systematic exploration.

## 2. Materials and Methods

### 2.1. Constructing Multiple Prokaryotic Expression Plasmids for SEM Protein and S178A Mutant

SignalP 4.1 [22] predicted an N-terminal signal peptide in SEM, consistent with secretion. The wild-type *sem* gene (GenBank: ADC37996) was PCR-amplified from *S. aureus* SA003 genomic DNA, isolated using a commercial kit. The mutant *sem* gene (S178A) was commercially synthesized. Both genes, lacking signal peptides (ΔNspSEMWT and ΔNspSEMS178A), were directionally cloned into pET-28a(+) vectors using introduced *Nde*I and *Xho*I sites.

To eliminate thrombin contamination risk, dedicated equipment and workspaces were strictly isolated. Structural comparison of ΔNspSEMS178A to wild-type ΔNspSEMWT required removal of the His_6×_-tag from His_6×_-TCS-ΔNspSEMS178A. For thrombin-free processing, a mutant *sem* gene (S178A) encoding an N-terminal His_6×_-tag (including *Nde*I site) followed by a SUMO-tag, based on pSmart I, was commercially synthesized (His_6×_-SUMO-ΔNspSEMS178A). This mutant *sem* gene (S178A) fragment also included a 3′ *Xho*I site with protective bases and was cloned into pSmart I using its native *Nde*I and *Xho*I sites. This strategy permits specific cleavage using SUMO protease, which recognizes the SUMO-tag’s C-terminal x-G-G-x motif and hydrolyzes the peptide bond after GG, releasing target ΔNspSEMS178A protein. All recombinant plasmids were sequence-verified before transformation into *E. coli* Rosetta (DE3) for expression.

### 2.2. Prokaryotic Expression and Purification of SEM and SEM (S178A) Mutant Fusion Protein

*E. coli* Rosetta (DE3) strains harboring pET-28a(+) or pSmart I recombinant plasmids were grown overnight at 37 °C in Luria–Bertani (LB) medium containing 30 μg/mL kanamycin or 100 μg/mL ampicillin, respectively. Following a 1:100 dilution into fresh medium, cultures were grown to OD_600nm_ ≈ 0.6. Protein expression was induced with 0.5 mmol/L isopropyl *β*-D-1-thiogalactopyranoside (IPTG) for 4–12 h. Cells were harvested by centrifugation (4000× *g*, 10 min, 4 °C), washed and resuspended in 20 mL buffer A (50 mmol/L Tris-HCl, 500 mmol/L NaCl, pH 8.0). After sonication and centrifugation (10,000× *g*, 20 min, 4 °C), the supernatant was subjected to IMAC. Contaminants were removed with buffer A containing 50 mmol/L imidazole, and His_6×_-tagged fusion proteins (His_6×_-TCS-ΔNspSEM_WT_, His_6×_-TCS-ΔNspSEM_S178A_ and His_6×_-SUMO-ΔNspSEM_S178A_) were eluted using buffer A with 500 mmol/L imidazole. Protein purity was confirmed by SDS-PAGE, with storage at 4 °C (short-term) or −80 °C (long-term).

Prior to SE-HPLC and TLA assays (Section 2.8), the buffer for His_6×_-TCS-ΔNspSEM_WT_ was exchanged to buffer B (1 mmol/L Na_2_HPO_4_-NaH_2_PO_4_, 500 mmol/L NaCl, pH 7.4) before ultrasonication. The high salt concentration in buffer B suppressed TLA activity to prevent unintended cleavage during processing; activity was restored by centrifugal ultrafiltration and desalting (Section 2.3).

To confirm TLA loss in the catalytic triad mutant His_6×_-TCS-ΔNspSEM_S178A_, the purified protein in buffer A underwent identical centrifugal ultrafiltration desalting as the wild type (Section 2.3). This exchanged the buffer to low-sodium buffer C (1 mmol/L Na_2_HPO_4_-NaH_2_PO_4_, pH 7.4; [NaCl] < 1 mmol/L). Desalted samples equilibrated at 4 °C for varying durations were analyzed by SDS-PAGE (Section 2.3).

For His_6×_-SUMO-ΔNspSEM_S178A_, imidazole was removed by centrifugal ultrafiltration after IMAC purification, and the buffer exchanged to facilitate SUMO protease cleavage. Following overnight digestion at 4 °C with SUMO protease, the mixture was applied to Ni-NTA resin. The His_6×_-SUMO tag bound to the resin in the absence of imidazole, while the cleaved, untagged ΔNspSEM_S178A_ flowed through. The resin was regenerated with 500 mmol/L imidazole to remove the bound tag. Purified ΔNspSEM_S178A_ was desalted into NaCl-free buffer C via centrifugal ultrafiltration (Section 2.3) for subsequent circular dichroism (CD) and intrinsic fluorescence emission spectroscopy analyses (Section 2.9).

### 2.3. Desalting and Exchange of the Fusion Protein Buffer System via Centrifugal Ultrafiltration

For removal of the N-terminal His_6×_-tag, the purified protein was desalted via centrifugal ultrafiltration (Amicon^®^ Ultra Centrifugal Filter, manufactured by Merck Ltd, Chengdu, Sichuan Province, China, 10 kDa MWCO, sample volume 15 mL) into buffer C. The three-stage ultrafiltration protocol comprised: (I) concentrating the 15 mL Ni-affinity purified sample to ≤1.5 mL by refrigerated centrifugation (4 °C, 4000~6000× *g*); (II) equilibrating the buffer by diluting to 15 mL with buffer C followed by re-concentration; (III) repeating Stage II three times to reduce NaCl and imidazole concentrations to ≤1 mmol/L, thereby equilibrating the His_6×_-TCS-ΔNspSEM_WT_ and His_6×_-TCS-ΔNspSEM_S178A_ fusion protein in fresh buffer C. The buffer-exchanged sample (day 0) and aliquots collected at intervals (0.5, 1, 2, 3, 6, 10 days) during 4 °C storage were analyzed by SDS-PAGE. All samples were immediately mixed with SDS/*β*-mercaptoethanol, heat-denatured (100 °C, 10 min), flash-frozen in liquid N_2_, and stored at −80 °C until analysis.

When subsequent gel filtration (Section 2.6) and TLA measurement with S-2238 (Section 2.8) was required, buffer B (500 mmol/L NaCl) replaced buffer C during ultrafiltration. This substitution eliminated imidazole while maintaining ionic strength, preventing spontaneous R–G peptide bond hydrolysis in the TCS of His_6×_-TCS-ΔNspSEM_WT_ under low-salt conditions.

### 2.4. Analysis of Fusion Protein Primary Structure by Mass Spectrometry (MS) and N-Terminal Sequencing

Protein samples were resolved by SDS-PAGE. The target band was excised from the gel using a sterile blade and collected in a 1.5 mL microcentrifuge tube. Gel fragments were hydrated with ultrapure water to prevent dehydration prior to submission for mass spectrometry analysis (Beijing Protein Innovation Co., Ltd., Beijing, China, or Sangon Biotech (Shanghai) Co., Ltd., Shanghai, China). For N-terminal sequencing, the corresponding gel fragment was similarly processed and submitted to Sangon Biotech (Shanghai, China) Co., Ltd. (Shanghai, China) with hydration maintenance. Technical replicates were electrophoresed in parallel, with target bands digitally captured using a gel imaging system. Reference images were shared with service providers to confirm band positional accuracy.

### 2.5. Calibration of Gel Filtration Column via Protein Markers Using HPLC

A gel filtration protein molecular weight marker kit (12-200 kDa, Sigma-Aldrich^®^ catalog number: MWFG200, Jakarta Timur, Indonesia) was analyzed using an Agilent^®^ 1100 Series HPLC system (manufactured by Agilent Technologies, Santa Clara, CA, USA) equipped with a Waters™ XBridge Protein BEH SEC column (7.8 × 300 mm, 3.5 μm; manufactured by Waters™, Milford, MA, USA), autosampler, and variable wavelength detector (VWD). Isocratic elution employed buffer B at 0.86 mL/min with UV detection at 280 nm. Protein markers and blue dextran were prepared in buffer B: cytochrome C (horse heart, 12.4 kDa, 2 mg/mL), carbonic anhydrase (bovine erythrocytes, 29 kDa, 3 mg/mL), bovine serum albumin (66 kDa, 10 mg/mL), alcohol dehydrogenase (yeast, 150 kDa, 5 mg/mL), *β*-amylase (sweet potato, 200 kDa, 4 mg/mL), and blue dextran (2000 kDa, 2 mg/mL). The mixture (10 μL injection volume) was chromatographed at 25 °C. Elution volumns (V_e_) were determined from injection to peak maximum. The void volume (V_0_) was similarly determined using blue dextran. Given the packed bed total volume (V_t_) as 7 mL (column manual), apparent partition coefficients (K_av_) were calculated as K_av_ = (V_e_ − V_0_)/(V_t_ − V_0_). A calibration curve was generated by plotting log-transformed molecular weights versus corresponding K_av_ values.

### 2.6. SE-HPLC Purification of the Ni-Affinity-Purified His_6×_-TCS-ΔNspSEM_WT_ Fusion Protein

To eliminate Tris-base interference in ultraviolet (UV) spectra (Section 2.2), proteins from IPTG-induced cells were extracted via sonication in buffer B. Crude lysate aliquots were analyzed for His_6×_-TCS-ΔNspSEM_WT_ fusion protein expression by SDS-PAGE with coomassie/silver staining, while TLA was assessed chromogenically (Section 2.8). Ni-affinity purification followed Section 2.2 with modifications: elution employed 500 mmol/L imidazole in buffer B. Eluate purity was verified by both staining methods. Before TLA assessment, imidazole was removed by centrifugal ultrafiltration (Section 2.3) with Buffer B exchange. For size-based purification, 100 μL Ni-eluted protein was loaded directly onto the SE- chromatographic column under molecular weight determination conditions (Section 2.5). Chromatographic peaks were fractionated and analyzed via SDS-PAGE, silver staining and TLA. Notably, SE-HPLC separated macromolecular proteins from low-molecular-weight imidazole, eliminating dialysis requirements. Thus, centrifugal ultrafiltration dialysis was unnecessary before chromatographic loading. To ensure chromatographic purity, 100 μL fractions from the initial SE-HPLC run were subjected to two subsequent purification rounds under identical conditions, with collected fractions retained for downstream analyses.

### 2.7. Bradford-Based Quantification of His_6×_-TCS-ΔNspSEM_WT_ Fusion Protein

Protein concentrations in the crude extract (prepared by sonication), Ni-affinity chromatography eluent, and gel filtration eluent were quantified using the Bradford protein assay kit (TIANGEN BIOTECH, Beijing, Co., Ltd.; catalog number: PA102, Beijing, China), following the manufacturer’s instructions. A bovine serum albumin (BSA) standard curve was generated for total protein quantification.

### 2.8. Chromogenic Analysis of TLA in His_6×_-TCS-ΔNspSEM_WT_ Fusion Protein

The chromogenic substrate S-2238 (H-D-Phe-Pip-Arg-pNA 2HCl) incorporates a *p*-nitroaniline (*p*NA) leaving group attached via an amide bond. Thrombin recognizes the arginine residue and hydrolyzes this bond, releasing *p*NA, which absorbs at 405 nm. Activity quantification relies on continuous kinetic monitoring of absorbance at 405 nm. To evaluate TLA in recombinant His_6×_-TCS-ΔNspSEM_WT_ protein, positive and negative controls were implemented. Positive controls: Pre-incubated (37 °C, 15 min) aliquots of bovine thrombin stock (1 U/mL in buffer C with 150 mmol/L NaCl; 10 µL), S-2238 (3 mmol/L in buffer C with 150 mmol/L NaCl; 10 µL), and buffer C (150 mmol/L NaCl; 150 µL) were sequentially combined in a 96-well plate in the order buffer → substrate → enzyme, yielding a final volume of 170 µL. Reactions were initiated upon mixing, and hydrolysis rates were determined from the linear slope of absorbance (405 nm) versus time. Negative controls: Thrombin was replaced with an equal volume of buffer C (150 mmol/L NaCl) to quantify non-enzymatic hydrolysis. These controls confirmed serine protease-responsive detection (positive) and minimized background hydrolysis during incubation (negative), enabling precise enzymatic quantification. TLA in recombinant His_6×_-TCS-ΔNspSEM_WT_ was confirmed by time-dependent increases in 405 nm absorbance, consistent with positive controls. Activity was assessed at three purification stages: (1) Crude *E. coli* lysate: Lysate (51 µL; 500 mmol/L NaCl), NaCl-free S-2238 (10 µL; 3 mmol/L), and NaCl-free buffer C (109 µL) were pre-incubated (37 °C, 15 min) and mixed (final NaCl: 150 mmol/L). Activity was quantified as for positive controls (triplicate measurements). (2) IMAC-purified protein: Protein (51 µL; 500 mmol/L NaCl post-ultrafiltration), NaCl-free S-2238 (10 µL), and NaCl-free buffer C (109 µL) were processed identically to (1). (3) SE-HPLC-purified protein: Analyzed as in (2).

### 2.9. Characterization of Recombinant Wild-Type ΔNspSEM_WT_ and Mutant ΔNspSEM_S178A_ Proteins by CD and Intrinsic Fluorescence Emission Spectroscopy

CD measurements were performed using an AVIV Model 400 spectropolarimeter (Aviv Biomedical, Inc., Lakewood, NJ, USA) equipped with a thermoelectrically controlled cell holder. Protein samples (ΔNspSEM_WT_ and ΔNspSEM_S178A_, 0.2 mg/mL, 25 °C) in 0.5 mm pathlength quartz cuvettes were analyzed. Far-UV spectra (190~260 nm) were acquired under nitrogen purge with a 1 nm bandwidth, 50 nm/min scan rate, 1 nm data pitch, and signal-averaged over three scans. All spectra underwent sequential processing: (i) solvent baseline subtraction using matched buffer controls and (ii) conversion to mean residue ellipticity ([*θ*]_MRW_, deg·cm^2^·dmol^−1^) using [*θ*]_MRW_ = (*θ*_obs_ × MRW)/10 × *l* × *c*, where *l* is pathlength (cm), *c* is concentration (mg/mL), MRW is mean residue weight (g/mol), and *θ*_obs_ is observed ellipticity (mdeg).

Fluorescence emission spectra were recorded on a Hitachi F-4500 spectrofluorometer (Hitachi High-Tech, Tokyo, Japan) equipped with a thermostated water circulator. Spectra (300~400 nm) were acquired at an excitation wavelength of 295 nm and averaged over triplicate scans. Fluorescence intensities were corrected sequentially: (i) solvent blank subtraction, (ii) inner filter effect correction using: F_cor_ = F_obs_ × e^(Aex+Aem)/2^, where F_cor_ and F_obs_ are corrected and observed intensities, respectively, and A_ex_ and A_em_ are absorbances at excitation and emission wavelengths, and (iii) normalization to the initial intensity at 25 °C to eliminate thermal quenching effects. All reported fluorescence intensities are corrected values.

### 2.10. Preparation of the TLA Tag-Free ΔNspSEM_WT_ Protein and Its Complex with Substrate His_6×_-TCS-ΔNspSEM_WT_ for Subsequent MD Simulations

We employed AlphaFold 3 [23] to generate reliable structural models of the His_6×_-TCS-ΔNspSEM_WT_ fusion protein and tag-free ΔNspSEM_WT_ protein. These predicted structures served as initial coordinates for MD simulations. Following 40 ns of equilibration (confirmed by stabilized RMSD), we extended simulations by 10 ns and randomly selected 50 conformations per variant from the final trajectory. Using Packmol 17.333 [24], we constructed 50 enzyme–substrate complexes by constraining the distance between S178-OG6543 (ΔNspSEM_WT_) and R17-C251 (His_6×_-TCS-ΔNspSEM_WT_) to 2–6 Å, consistent with serine protease catalytic geometry. To address Packmol’s rigid-body limitation, we applied five heating–cooling cycles (0–350 K at 350 K/ns) for conformational refinement. Fifty annealed complexes exhibiting 3–4 Å catalytic spacing between S178-OG6543 and R17-C251 were randomly selected as initial MD structures (Section 2.10).

### 2.11. MD Simulations of Enzyme (ΔNspSEM_WT_) and Substrate (His_6×_-TCS-ΔNspSEM_WT_) Complex

All simulations used GROMACS 2019.5 [25] with the AMBER14SB force field [26] under periodic boundary conditions. The ΔNspSEM_WT_ and His_6×_-TCS-ΔNspSEM_WT_ complex was solvated in a 113.18 Å cubic box (maintaining ≥ 14 Å protein-wall clearance) with TIP3P water. Systems underwent energy minimization via steepest descent (convergence: 1000 kJ/(mol·nm); step size: 0.01 nm), followed by 30 ps position-restrained equilibration (protein fixed, solvent relaxed). Production simulations (150 ns) maintained 298.15 K using the Berendsen thermostat (λ = 0.5 ps; solute-solvent decoupled) and 1 bar pressure via the Parrinello–Rahman barostat (τ = 1.0 ps; isothermal compressibility = 4.5 × 10^−5^ bar^−1^). Electrostatics employed 15 Å short-range cutoffs and Particle Mesh Ewald (PME) for long-range interactions, with 1.2 Å grids and fourth-order splines.

## 3. Results and Discussion

### 3.1. Desalting-Induced Autocatalytic Activation of His_6×_-TCS-ΔNspSEM_WT_ and Proteolytic Degradation Observed by SDS-PAGE

Due to the absence of SEM crystal structures, we predicted its 3D model using AlphaFold 3 (global pLDDT score: 93.51, pTM score: 0.93) and Swiss-Model [27] (Figure 1a). Structural alignment yielded a Cα-RMSD of 3.6 Å, indicating minimal backbone divergence. Like all SEs, SEM exhibits three conserved motifs: an N-terminal OB-fold domain with *β*-barrel and exposed cysteine (Cys_93_) for disulfide bonding; a C-terminal *β*-grasp domain containing antiparallel *β*-strands and a Zn^2+^-binding site [28] coordinated by histidine 190, histidine 228, and aspartic acid 230 (H_190_/H_228_/D_230_); and interconnected by a central diagonal *α*-helix. The His_6×_-TCS-ΔNspSEM_WT_ fusion protein was purified via IMAC in buffer A (Figure 1b, lane 1), exhibiting an apparent mass of ~27 kDa consistent with its calculated mass. Minor dimeric/tetrameric species indicated residual aggregation. Prior to characterization, buffer exchange into a low-salt buffer C (<1 mmol/L NaCl) via centrifugal ultrafiltration (4 °C, Section 2.3) eliminated imidazole and optimized conditions for thrombin cleavage at the TCS R–G bond. Unexpectedly, SDS-PAGE revealed a sharp novel band below the parental protein after buffer exchange (Figure 1b, lane 2), despite no protease addition. The sharp bands without smearing strongly suggest site-specific autocatalysis in this Ni-affinity purified protein. This unprecedented self-cleavage under low ionic strength implies intrinsic protease-like activity in the His_6×_-TCS-ΔNspSEM_WT_ fusion construct.

This finding intrigued us, as SEM protease-like activity remained unreported since its initial characterization by S Jarraud et al. in 2001 [7]. To preclude artifacts from exogenous proteases, we rigorously followed protocols (Section 2.2 and Section 2.3) using fresh reagents, a new plasmid kit, commercial sequencing, and newly transformed *E. coli*. Lysate was purified/desalted with new Ni-NTA media and Amicon^®^ Ultra filters through repeated exchanges into salt-free buffer C (<1 mmol/L NaCl). Samples included: (1) “0 day” (immediately post-exchange), and (2) aliquots stored statically at 4 °C for 0.5–10 days. Salt-free samples and IMAC-purified high-salt (500 mmol/L NaCl) controls underwent SDS-PAGE. Figure 1c shows desalted day 0 samples (lanes 0 d) exhibiting two bands (faint, intense), indicating autocatalytic conversion. The degradation product intensified by day 0.5, stabilizing through day 10, while the parental 27-kDa band disappeared by day 1. A minor 16-kDa band emerged at day 0, intensified slightly by day 3, and persisted. Site-specific proteolysis was confirmed by absent nonspecific degradation. The observed 2-kDa shift corresponds to the theoretical His_6×_-tag mass (1.9 kDa) from pET-28a(+). Thus, we propose that His_6×_-TCS-ΔNspSEM_WT_ autocatalytically cleaves the TCS sequence (LVPR↓GS) at R–G, liberating the His_6×_-tag. To validate this, the novel 25-kDa band (Figure 1c, lane 1 d, black arrow) was excised for MS and N-terminal sequencing. Figure 1d–g presents MS results: peptide scores > 28 were significant (*p* < 0.05), indicating confident matches between MS/MS spectra and Mascot-searched peptide sequences (Figure 1d). One protein showed a strongly significant match (score 77,111; Figure 1e). High sequence coverage (87%; Figure 1f) confirms identification. Moreover, the sequence aligns with signal-peptide-removed SEM (GenBank: D0K667), with 96.3% coverage. Crucially, no His_6×_-tag peptides were detected by MS, confirming His_6×_-tag removal during desalting and storing at 4 °C. N-terminal sequencing identified GSXMDVGVLN (Appendix A). Position 3 (X) yielded undetectable PTH-amino acids; given PTH-cysteine exclusion from Edman chemistry, X was assigned as cysteine (C). Our design intended thrombin cleavage to generate N-terminal GSHMDVGVLN (tetrapeptide GSHM from vector; hexapeptide DVGVLN from ΔNspSEM’s native N-terminus). The sole discrepancy was position 3 (C instead of H), which does not invalidate the demonstrated TLA in the His_6×_-TCS-ΔNspSEM_WT_ fusion protein cleaving at R–G bond, releasing ΔNspSEM with GSXMDVGVLN N-terminus.

In summary, the 25-kDa band confirms specific R–G bond proteolysis in His_6×_-TCS-ΔNspSEM_WT_, demonstrating intrinsic TLA. This recombinant protein exhibits desalting-activated, NaCl-inhibited TLA that specifically cleaves the TCS R–G peptide bond.

### 3.2. Two-Step Purification via IMAC Followed by SE-HPLC and S-2238-Based Chromogenic Detection of His_6×_-TCS-ΔNspSEM_WT_

Single-step IMAC may incompletely remove proteolytic enzymes from bacterial lysates. We therefore employed sequential IMAC/SE-HPLC purification and evaluated TLA of the purified His_6×_-TCS-ΔNspSEM_WT_ fusion protein using S-2238 at different stages. First, the SE-chromatographic column was calibrated with protein markers (Section 2.5). Figure 2a shows five well-resolved symmetrical peaks corresponding to these markers. Only blue dextran (for V_0_ determination) exhibited fronting peak, reflecting inherent molecular heterogeneity. A calibration curve (Figure 2e) plotting log molecular weights against K_av_ established an SE-HPLC system suitable for target protein separation and apparent molecular weight determination. Prior to assessing TLA of the fusion protein, we verified the detection system reliability using a positive control (Section 2.8). Figure 2h shows a clear linear increase in ΔA_405 nm_ (+0.183) over 20 h at 37 °C, demonstrating that S-2238, efficiently hydrolyzed by thrombin to release *p*NA, shows potential as a suitable substrate for the TLA-exhibiting protein. Negative controls confirmed S-2238 chemical stability, showing minimal spontaneous hydrolysis (ΔA_405 nm_ ≤ 0.003 over 20 h at 37 °C, Figure 2h, blue points). Consequently, control experiments validated the S-2238-based assay for accurate TLA quantification. Following the Section 2.8 protocol, we quantified TLA of the His_6×_-TCS-ΔNspSEM_WT_ fusion protein using this validated assay. Soluble lysates prepared by sonication in buffer B (Section 2.2) exhibited significant TLA (A_405 nm_ = 0.440 after 20 h at 37 °C, Figure 2i, red dots), confirming the protein’s ability to hydrolyze S-2238 and release *p*NA. TLA was thus quantified by measuring ΔA_405 nm_ over time. Sonicated lysates showed an elevated baseline (A_405 nm_ = 0.150) due to unidentified pigments, yet still demonstrated a substantial ΔA_405 nm_ increase (+0.290). SDS-PAGE confirmed efficient fusion protein expression at the predicted molecular weight (Figure 2f,g, lane 1).

However, crude lysates obtained by ultrasonic disruption contained significant host-derived contaminants. We therefore employed IMAC as the initial purification step, following the protocol established in Section 2.2. SDS-PAGE analysis of the IMAC-purified fusion protein (Figure 2f,g, lane 2) revealed a prominent monomer band (27 kDa), a faint dimer band (54 kDa), and diffuse lower molecular weight bands, confirming efficient purification. Notably, using phosphate buffer instead of Tris-HCl for sonication and IMAC (Section 2.2) demonstrated that spontaneous hydrolysis at the R–G bond within the TCS motif was buffer-independent, but critically suppressed by high ionic strength (500 mmol/L NaCl). Following imidazole removal by centrifugal ultrafiltration, we quantified TLA of the IMAC-purified protein (Section 2.3 and Section 2.8). The ΔA_405nm_ (+0.178) increase after 20 h (Figure 2i, green circles) was comparable to the thrombin positive control rate (Figure 2h, violet circles), demonstrating that IMAC enhanced purity while preserving TLA. This IMAC step yielded an electrophoretically pure, TLA-active fusion protein His_6×_-TCS-ΔNspSEM_WT_. To eliminate low-abundance contaminants below 27 kDa (Figure 2f,g, lane 2), we performed a second purification step using SE-HPLC (Section 2.6). Analysis revealed four symmetrical peaks at 9.173, 9.665, 10.935, and 11.392 min (Figure 2b). Calibration identified species of 83.5, 56.8, 21.0, and 14.7 kDa. Correlating with the theoretical mass (27.136 kDa) and SDS-PAGE migration (27 kDa), these data indicate a predominant dimeric conformation (56.8 kDa at 9.665 min) in buffer B, alongside minor trimer (83.5 kDa), monomer (21.0 kDa), and proteolytic fragment (14.7 kDa) populations. The dimeric fractions eluting at 9.665 min (56.8 kDa) were pooled. SDS-PAGE analysis (Figure 2f,g, lane 3) showed dissociation during denaturation to a prominent 27 kDa monomer band. Both coomassie blue and silver staining indicated high purity, with only trace dimer remaining. TLA analysis of this fraction (Section 2.8) revealed a ΔA_405nm_ increase of +0.037 (Figure 2i, green dots), exceeding the negative control but below the positive control, confirming detectable TLA.

The 9.665 min fraction from the first SE-HPLC round was further purified by a second SE-HPLC separation. Re-analysis confirmed the chromatographically pure fusion protein dimer eluted predominantly at 9.641 min, accompanied by a minor monomer at 10.954 min (Figure 2c). SDS-PAGE of the 9.641-min fraction (Figure 2f,g, lane 4) matched the profile from the first SE-HPLC dimer purification (Figure 2f,g, lane 3). A third SE-HPLC cycle of this dimer fraction (9.641 min) yielded a major peak at 9.656 min and a minor peak at 10.951 min, with elution profiles superimposing on previous cycles. The absence of trimer and fragments during secondary and tertiary SE-HPLC, coupled with retention time consistency, demonstrates a dimer-monomer equilibrium in buffer B, confirming the His_6×_-TCS-ΔNspSEM_WT_ fusion protein dimer as the predominant conformation. In summary, chromatographically pure His_6×_-TCS-ΔNspSEM_WT_ fusion protein was obtained via sequential IMAC and SE-HPLC. This protein existed predominantly as dimers under high-salt conditions, wherein its TLA was suppressed. Reducing salt concentration restored TLA, enabling slow hydrolysis of S-2238.

Table 1 outlines the purification scheme for bacterially expressed fusion protein His_6×_-TCS-ΔNspSEM_WT_. Sequential IMAC/SEC-HPLC achieved 7.059-fold TLA purification from crude lysates, yielding chromatographically pure protein with 0.360 U/mg specific activity and 0.09% recovery. This measurable yet modest purification confirms successful isolation of functional His_6×_-TCS-ΔNspSEM_WT_ fusion protein. The limited purification efficiency and catalytic parameters suggest either suboptimal S-2238 recognition by this fusion protein or potential inhibition of S-2238 proteolysis under physiological buffer conditions. Collectively, we demonstrate successful expression, purification and structural characterization of His_6×_-TCS-ΔNspSEM_WT_ fusion protein, which predominantly forms NaCl-stabilized dimers (500 mmol/L NaCl).

### 3.3. Impact of Ionic Strength, PH and Substrate on TLA in His_6×_-TCS-ΔNspSEM_WT_ Fusion Protein

Following IMAC purification of His_6×_-TCS-ΔNspSEM_WT_ via imidazole elution, buffer B (500 mmol/L NaCl) replaced the imidazole/NaCl-containing buffer using centrifugal ultrafiltration (Section 2.3). The imidazole-free fusion protein in high-salt buffer B (500 mmol/L NaCl) was incubated at 37 °C for 10 days. Aliquots collected daily (days 1~6, 8, 10) showed no autoproteolysis by SDS-PAGE, confirming full-length integrity under conditions optimal for thrombin activity (Figure 3a). Parallel samples in NaCl-supplemented buffer B (lacking imidazole) were stored at 37 °C for identical durations (1~6, 8, 10 days), then desalted into NaCl-free buffer C via centrifugal ultrafiltration (Section 2.3) until [NaCl] < 1 mmol/L. Subsequently, all desalted samples were subjected to incubation at 37 °C for 6 h prior to SDS-PAGE analysis. SDS-PAGE revealed that 6-h incubation at 37 °C induced autoproteolysis, releasing ΔNspSEM_WT_ (Figure 3b). These results demonstrate that TLA activation requires low ionic strength and is suppressed in high-salt environments. The observed TLA suggests mechanistic parallels with thrombin’s serine proteolytic activity. AlphaFold 3 structural modeling (Figure 3f) identified the canonical catalytic triad (H_172_/S_178_/D_212_) [29] on the *β*-grasp domain surface. This integration of experimental and computational data demonstrates intrinsic serine protease activity. Consequently, pH-dependent regulation occurs through protonation/deprotonation of catalytic H_172_, explaining TLA suppression under both acidic (pH 2.0~5.5) and alkaline (pH 10.0~12.0) conditions. To examine pH effects, buffer exchange from buffer B to salt-free buffer C at various pH values was performed using IMAC-purified protein. Subsequent 24-h proteolytic reactions at 4 °C revealed maximal TLA activation in neutral-to-weakly-basic environments (pH 6.5~9.0). The fusion protein demonstrated TCS motif recognition and R–G peptide cleavage, enabling autonomous His_6×_-tag excision. This establishes signal peptide-deficient ΔNspSEM_WT_ as a promising enzymatic tool for His_6×_-tag removal in prokaryotic and eukaryotic systems. To evaluate substrate specificity, IMAC-purified His_6×_-TCS-ΔNspSEM_WT_ fusion protein underwent buffer exchange into salt-free buffer C to activate proteolysis. Time-course SDS-PAGE showed complete His_6×_-tag excision within 24 h at 4 °C (Figure 3e, Lane 3). Co-incubation of ΔNspSEM_WT_ with bovine serum albumin (BSA) or hen egg-white lysozyme (HEWL) (1:1 mass ratio, salt-free, 37 °C, 8 h) showed no detectable nonspecific degradation (Figure 3e, Lanes 5 and 8), with individual controls confirming component integrity (BSA-Lane 4; HEWL-Lane 7; ΔNspSEM_WT_-Lane 6). Cleavage strictly required the TCS motif (LVPR↓GS), with no detectable activity observed in motif-deficient substrates, demonstrating ΔNspSEM_WT_’s precision for engineered tag removal.

### 3.4. S178A Mutation in the Catalytic Triad Abolishes TLA of His_6×_-TCS-ΔNspSEM_WT_

In Section 3.2, purification of the His_6×_-TCS-ΔNspSEM_WT_ fusion protein via sequential IMAC and SE-HPLC revealed TLA using S-2238. While SDS-PAGE confirmed increasing purity throughout purification (Figure 2f,g), both final yield and TLA decreased significantly (Table 1). Quantitative analysis, however, demonstrated corresponding increases in specific activity and purification fold, indicating effective contaminant removal and target protein enrichment. Absolute purity (100%) remains impractical in any purification scheme. Therefore, potential contributions of trace endogenous *E. coli* serine protease contamination to the observed activity must be excluded. To definitively attribute TLA to the catalytic triad (H_172_/S_178_/D_212_) within ΔNspSEM_WT_’s *β*-grasp domain, we generated the S178A [30] mutant His_6×_-TCS-ΔNspSEM_S178A_. This mutant was expressed in *E. coli*, purified by IMAC, and analyzed by SDS-PAGE under identical conditions to the wild-type protein. IPTG induction yielded mutant protein with electrophoretic mobility identical to the wild-type (Figure 4a, Lane 2 uninduced and Lane 3 induced). Following large-scale expression and IMAC purification (Section 2.2), the mutant fusion protein was desalted by centrifugal ultrafiltration at 4 °C (Section 2.3). SDS-PAGE analysis of desalted samples taken at identical time points to wild-type (Section 2.3, Figure 1c, lanes 0 d~10 d vs. Figure 4a, lanes 4~10) revealed that the mutant protein, unlike the wild-type, undergoes no peptide-specific degradation. The electrophoretic band (Figure 4a, Lane 10, black arrow) from the desalted sample incubated at 4 °C for 10 days was excised and submitted for MS sequencing. As shown in Figure 4d, the peptide coverage obtained from MS analysis exactly matched the primary structure of our engineered mutant fusion protein His_6×_-TCS-ΔNspSEM_S178A_.

Notably, substituting the solvent-exposed, polar serine 178 within the *β*-grasp domain with hydrophobic alanine could invert the local microenvironment, potentially disrupting the catalytic triad’s 3D structure. Under extreme conditions, such a mutation might also alter secondary structure, compromising tertiary fold and destroying the active site. To rigorously exclude structural destabilization (secondary or tertiary) as the cause of TLA loss and eliminate potential thrombin contamination in the assay, we constructed the His_6×_-SUMO-ΔNspSEM_S178A_ expression plasmid (Section 2.1). Following prokaryotic expression and IMAC purification (Section 2.2), the His_6×_-SUMO tag was cleaved with SUMO protease, and the tag-free ΔNspSEM_S178A_ was isolated by reverse IMAC purification. Finally, CD and intrinsic fluorescence emission spectra of wild-type ΔNspSEM_WT_ and mutant ΔNspSEM_S178A_ were acquired at 25 °C (Section 2.9). Figure 4b,c shows nearly identical CD spectra (secondary structure) and intrinsic fluorescence spectra (local hydrophobicity around the *β*-grasp domain tryptophan) for both proteins. This demonstrates the mutant possesses secondary and tertiary structures indistinguishable from the wild-type at ambient temperature. Consequently, the loss of TLA in the ΔNspSEM_S178A_ mutant is conclusively attributed to disruption of the catalytic mechanism, not global structural perturbation.

In conclusion, comparative structural and functional analysis of the mutant and wild-type fusion proteins robustly confirms the intrinsic TLA of the SEM protein.

### 3.5. Probing Substrate Specificity and TLA in ΔNspSEM_WT_ via MD Simulations

Classical MD simulations provide atomic-level insights into enzymatic substrate-binding specificity. While protocols typically prioritize high-resolution crystal structures (≤2 Å), the absence of SEM crystal structure in the PDB necessitates alternative approaches. We generated initial models using AlphaFold 3 and homology modeling. Post-equilibration analyses revealed strong agreement between simulated secondary structures and circular dichroism data, complemented by computational solvent accessibility assessments of tryptophan residues correlating with fluorescence emission spectra [31]. These experimental validations confirm structural reliability.

Figure 5a shows equilibrated 150 ns simulations of ΔNspSEM_WT_ (His_6×_-tag-free) and its His_6×_-tagged substrate forming stable complexes, with C_α_–RMSD remaining converged over the final 110 ns (40~150 ns). Trajectory analysis identified a canonical catalytic triad (H_172_/S_178_/D_212_), with S_178_–OG_6543_ in the *β*-grasp domain acting as the nucleophile attacking R_17_–C_251_ in the TCS motif (Figure 5b,e). Critical S_178_–OG_6543_/R_17_–C_251_ distances fluctuated between 3~8 Å, with periodic approaches to 3~4 Å indicating catalytic competence (Figure 5b). Substrate recognition was stabilized by dual salt bridges (R_17_–D_212_/E_214_ ≤ 4 Å, Figure 5c,d) and thirteen hydrogen bonds (eight involving R_17_’s sidechain), with five additional hydrogen bonds enhancing specificity (Figure 5f). This configuration strategically positions the R_17_–G_18_ peptide bond near S_178_–OG_6543_ via electrostatic steering, establishing a “molecular guillotine” mechanism. The catalytic platform anchors R_17_ through D_212_/E_214_ while exposing S_178_–OG_6543_ as the cleavage element. At critical proximity (3~4 Å), this nucleophile severs the R_17_-G_18_ bond, releasing the His_6×_-tagged fragment from ΔNspSEM_WT_’s C-terminal domain. Collectively, S_178_–OG_6543_ acts as a precision catalytic element, selectively cleaving the R_17_–G_18_ bond within the *β*-grasp domain.

## 4. Conclusions

Based on experimental and simulation data, the surface-exposed residues H_172_/S_178_/D_212_ in SEM’s *β*-grasp domain constitute a canonical serine protease catalytic triad. Crucially, the N-terminal R_17_ of hydrolyzed R_17_–G_18_ peptide bonds anchors to SEM’s *β*-grasp surface via salt bridges and hydrogen bonds with D212/E214. This substrate fixation mechanism likely drove SEM’s evolutionary acquisition of TLA targeting R_17_–G_18_ bonds. Structural analyses integrating PDB and AlphaFold 3 predictions reveal five distinct evolutionary patterns within the staphylococcal enterotoxin (SE) superfamily: Type I (SEH/SEI/SEQ): preserves SEM’s catalytic triad and R_17_-recognition salt bridges. Type II (SEO/SEP/SES): retains triad but exhibits divergent substrate-binding interfaces. Type III (SET/SEY/SEY3/SEZ): evolved novel triads with reconfigured salt-bridged binding pockets. Type IV (SEA/SEN): retains triad but lacks R_17_-anchoring salt bridges. Type V (SEB/SEC/SED/SEE/SEG/SEJ/SEL/SEK/SEO2/SER/SEW/SEX): lost triad architecture, correlating with absent protease activity. These findings raise fundamental questions: why did nearly half of SEs evolve functional protease triads? What selective pressures drove specific enterotoxins to develop arginine-targeting pockets, refining TLA? Most critically, what pathogenic advantages do these activities confer? Addressing these knowledge gaps requires integrated structural phylogenetics, enzymatic assays, and host–pathogen studies. We hypothesize that SEs’ acquired protease/TLA activities mechanistically link to disseminated coagulation in severe *S. aureus* infections. While not exclusive contributors, TLA may cleave fibrinogen’s R–G bonds, generating fibrin monomers that polymerize into soft hydrogels. Crucially, fibrin-based soft hydrogel formation on bacterial surfaces may serve as a dual-functional immune evasion strategy: (1) establishing a physical barrier that masks surface antigenic heterogeneity, and (2) impairing phagocytic clearance by macrophages/neutrophils while attenuating lymphocyte-mediated attacks. This proposed mechanism would facilitate hematogenous dissemination of *S*. *aureus*, enabling systemic colonization of distant tissues and organs. Systematic investigations into this hypothesis could elucidate the pathophysiological significance of SEs’ enzymatic diversification during host invasion, potentially revealing novel therapeutic targets against complicated staphylococcal infections.

Inspired by salt-concentration-dependent TLA modulation (suppressed at ≥500 mM NaCl; activated upon desalting), we propose a heterologous expression system design featuring: target protein-TCS(LVPR↓GS)-His_6×_-SEM. Following IMAC purification and buffer exchange, desalted SEM would specifically cleave the R–G bond, releasing the target protein while retaining the His_6×_-tagged SEM for subsequent IMAC removal. This strategy would outperform thrombin-based methods by: (1) eliminating exogenous proteases; (2) enabling simultaneous tag removal and buffer optimization during 4 °C storage; (3) minimizing denaturation; and (4) enhancing product concentration by collecting flow-through, thus bypassing dilution-prone gel filtration. Collectively, this integrated purification-cleavage approach would reduce processing steps, minimize dilution, and streamline industrial-scale protein production, offering significant cost and scalability advantages.

## Figures and Tables

**Figure 1 biomolecules-15-01357-f001:**
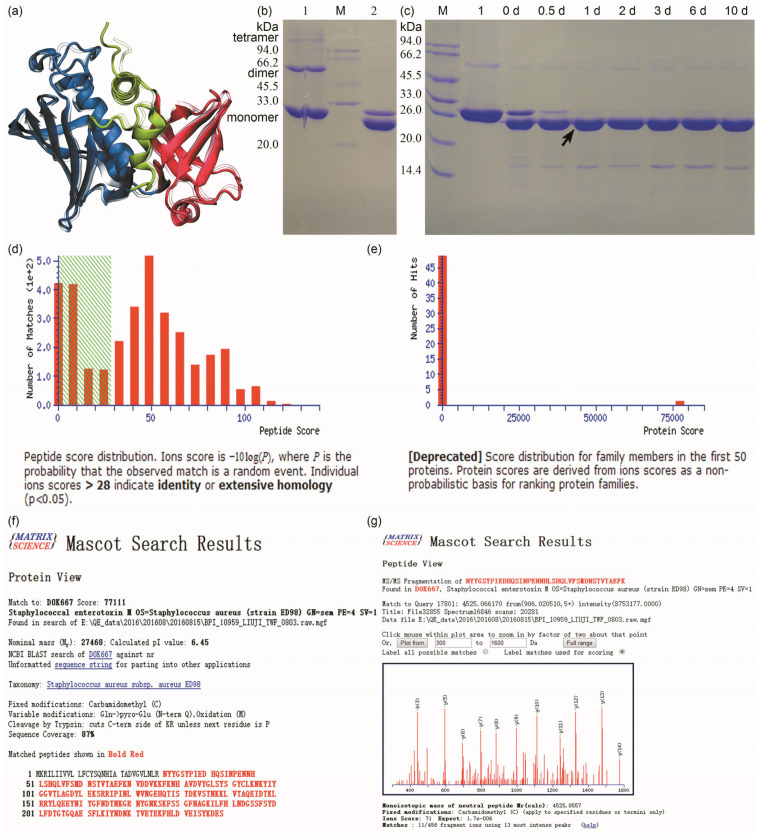
Structural prediction of SEM, IMAC purification of the His_6×_-TCS-ΔNspSEM_WT_ fusion protein, site-specific cleavage at R–G bonds within TCS motif and MS validation of the recombinant fusion protein. (**a**) SEM structure prediction using AlphaFold 3 (solid) and Swiss-Model homology modeling (transparent). (**b**) Unexpected proteolytic cleavage at the TCS R–G bond following ultrafiltration desalting. Lane 1: IMAC-purified His_6×_-TCS-ΔNspSEM_WT_; Lane M: molecular weight markers; Lane 2: Desalted fusion protein. (**c**) Time-course validation of TCS-specific cleavage. Lane M: Markers; Lane 1: IMAC-purified fusion protein; Lanes 3~9: Samples stored at 4 °C after desalting (0, 0.5, 1, 2, 3, 6, 10 days). (**d**) Peptide score distribution. (**e**) Protein score distribution. (**f**) Protein view. (**g**) Peptide view. Original images can be found in Appendix A.

**Figure 2 biomolecules-15-01357-f002:**
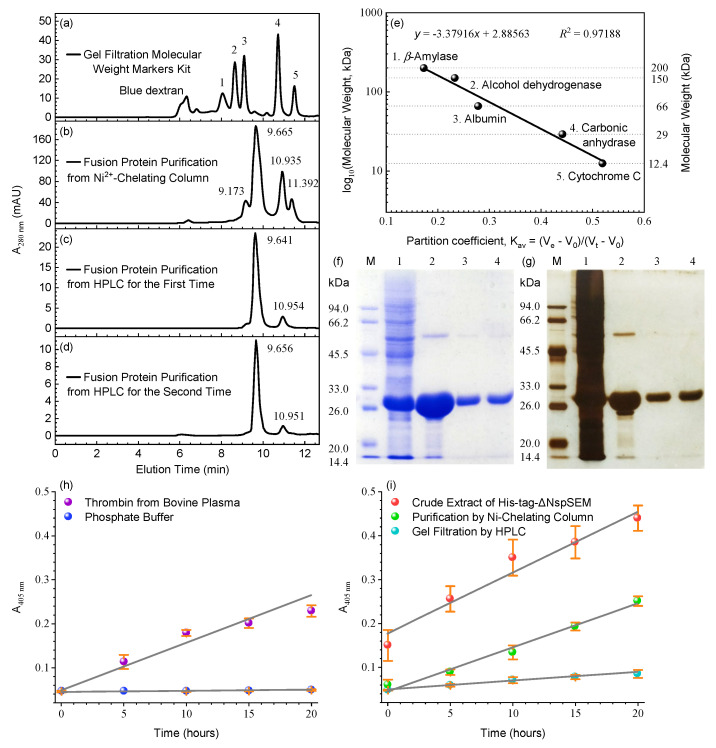
The His_6×_-TCS-ΔNspSEM_WT_ fusion protein was purified to chromatographic homogeneity by IMAC followed by SE-HPLC, with TLA quantification after each step. (**a**) SE-HPLC analysis of protein molecular weight markers according to Section 2.5. (**b**) The IMAC-purified fusion protein was further purified by SE-HPLC. (**c**) SE-HPLC purification of the 9.655-min fraction from (**b**). (**d**) SE-HPLC repurification of the 9.641-min fraction from (**c**). (**e**) SE-HPLC calibration curve using protein molecular weight markers. (**f**) SDS-PAGE (coomassie brilliant blue) of purified fusion protein. Lane M: Protein markers; Lane 1: Crude extract via ultrasonication; Lane 2: IMAC-purified fusion protein; Lane 3: Fractions (9.655 min) from the first round of SEC-HPLC purification; Land 4: Fractions (9.641 min) from the secondary round of SEC-HPLC purification. (**g**) Silver-stained SDS-PAGE of purified fusion protein. Lane M, Lane 1~3: Samples identical to (**f**). (**h**) Validation of the S-2238-based TLA assay using positive and negative controls. (**i**) TLA activity monitoring via S-2238 hydrolysis during purification. Original images can be found in Appendix A.

**Figure 3 biomolecules-15-01357-f003:**
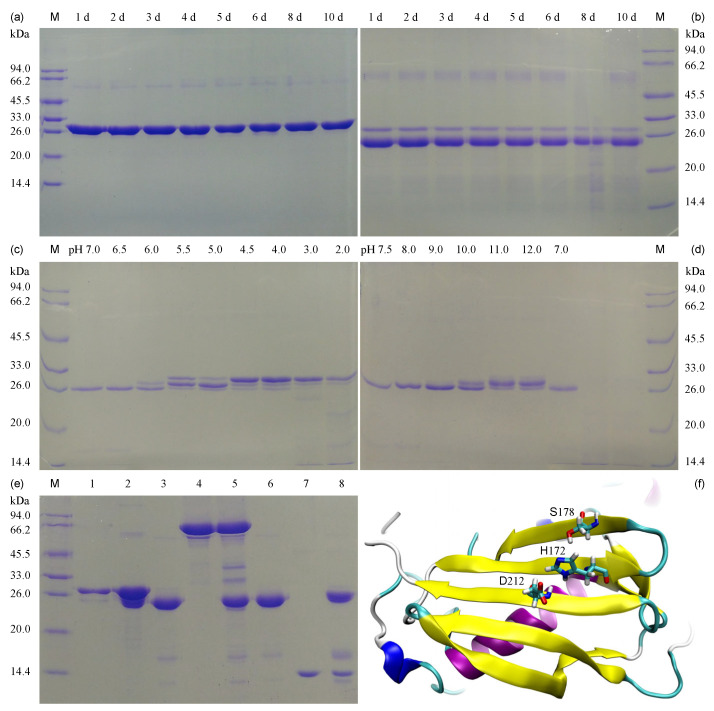
Effects of ionic strength, pH and TCS-free substrates on TLA activity in His_6×_-TCS-ΔNspSEM_WT_ fusion protein. (**a**) Fusion protein exhibits inhibited TLA under high-salt conditions. Lane M: protein marker. Lanes 1~6, 8, 10: daily aliquots collected at days 1~6, 8, 10. (**b**) Ultrafiltration desalting resumes TLA. Lanes 1~6, 8, 10: daily aliquots collected at days 1~6, 8, 10. Lane M: protein marker. (**c**) Neutral-to-acidic pH modulates TLA following centrifugal desalting. Lane M: protein marker. Lane pH: pH-controlled buffer exchange (2.0, 3.0, 4.0, 4.5, 5.0, 5.5, 6.0, 6.5 and 7.0) during centrifugal desalting of the fusion protein. (**d**) Neutral-to-alkaline pH regulates TLA in fusion protein. Lane pH: pH-controlled buffer exchange (7.5, 8.0, 9.0, 10.0, 11.0 and 12.0) during centrifugal desalting of the fusion protein. Lane M: protein marker. (**e**) TCS motif-deficient substrates (BSA, HEWL) modulate TLA in fusion protein. Lane 1: His_6×_-TCS-ΔNspSEM_WT_ fusion protein; Lane 2: Desalted His_6×_-TCS-ΔNspSEM_WT_ fusion protein; Lane 3: His_6×_-tag cleaved ΔNspSEM_WT_ protein; Lane 4: Bovine serum albumin (BSA); Lane 5: ΔNspSEM_WT_ + BSA mixture; Lane 6: Same as lane 3; Lane 7: Hen egg-white lysozyme; Lane 8: ΔNspSEMWT + lysozyme mixture. (**f**) The catalytic triad (histidine-172: H_172_, aspartic Acid-212: D_212_, serine-178: S_178_) predicted by AlphaFold 3 n *β*-grasp domain surface of ΔNspSEM_WT_ recombinant protein. Original images can be found in Appendix A.

**Figure 4 biomolecules-15-01357-f004:**
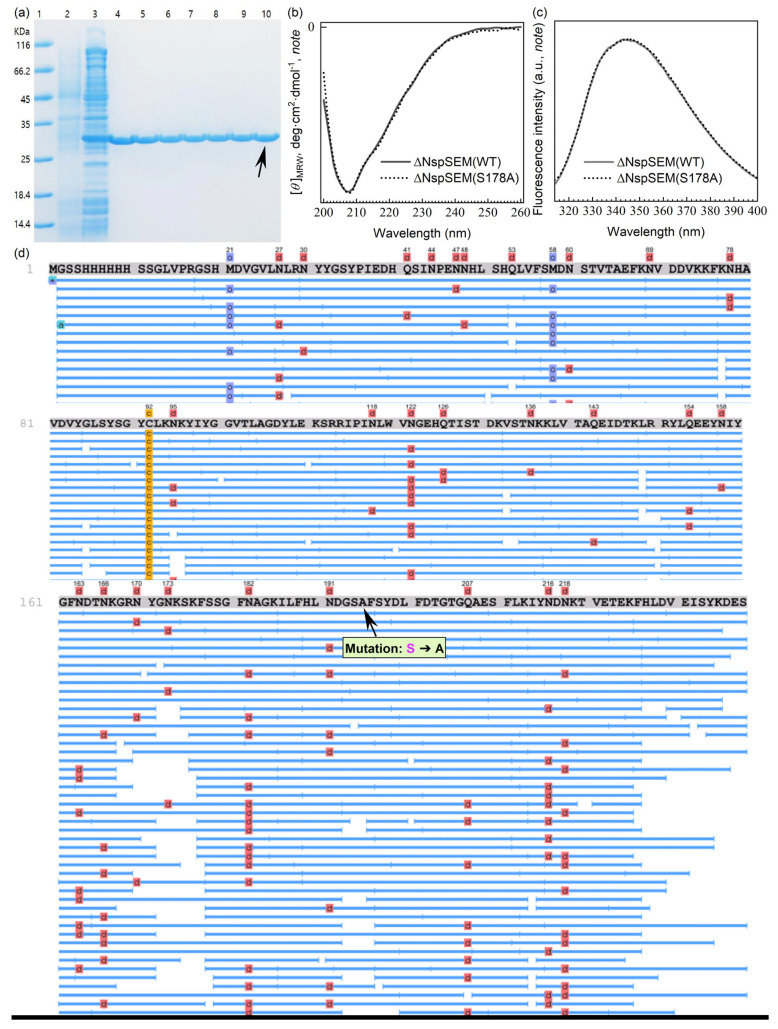
Validation of the loss of TLA for specific hydrolysis at R-G peptide bonds within the TCS sequence by the mutant His_6×_-TCS-ΔNspSEMS178A fusion protein. (**a**) Ultrafiltration and desalting of IMAC-purified mutant fusion protein. Lane 1, protein molecular weight markers; Lane 2, unin-duced *E. coli* lysate; Lane 3, induced *E. coli* lysate; Lanes 4~10: samples of the electrophoretically purified mutant protein after centrifugal ultrafiltration desalting, stored statically at 4 °C and collected at 0, 0.5, 1, 2, 3, 6, and 10 days. (**b**,**c**) CD spectra and fluorescence intensity (*note*: a.u., arbitrary units,) of His_6×_-tag-free wild-type ΔNspSEMwt and mutant ΔNspSEMS178A protein. *Note*. Y-values omitted to highlight spectral intensity signals at different wavelengths. (**d**) The electrophoretic band (Figure 4a, Lane 10, black arrow) was excised and analyzed by mass spectrometry for primary structure determination. The “Mutation: S→A” indicated by the black arrow shows the site where the serine residue was replaced by alanine in the catalytic triad (H172/S178/D212) of the SEM protein.

**Figure 5 biomolecules-15-01357-f005:**
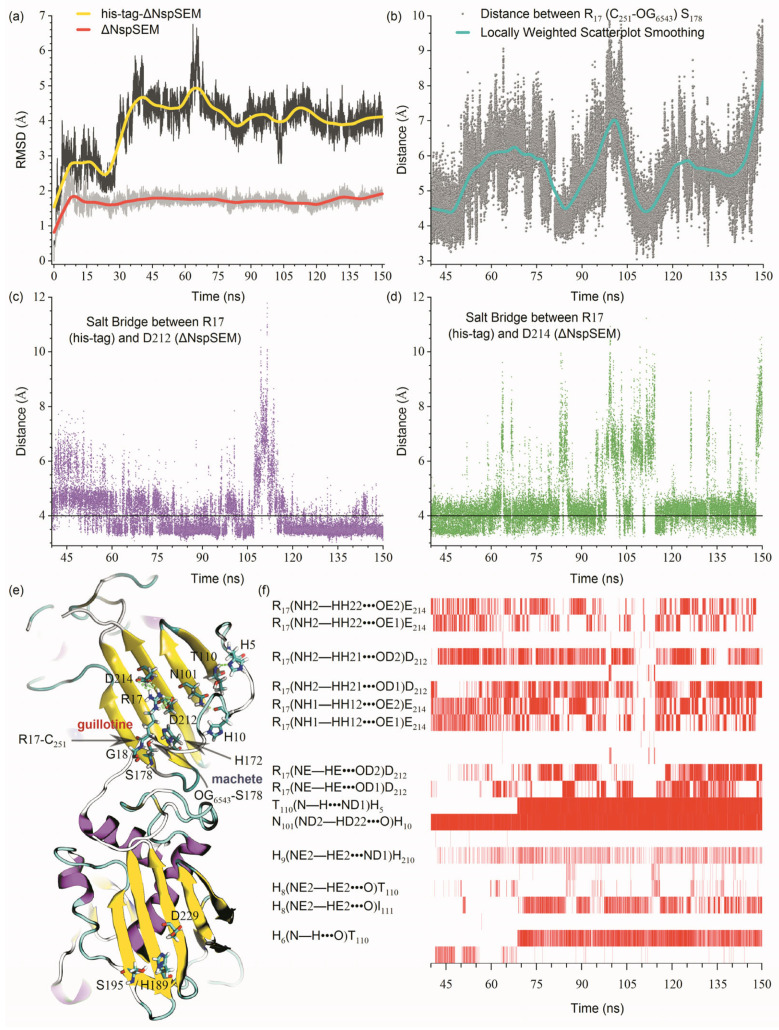
MD simulations probe the molecular mechanism of His_6×_-TCS-ΔNspSEM_WT_ substrate binding to the ΔNspSEM_WT_ active center. (**a**) Time evolution of RMSD for His_6×_-TCS-ΔNspSEM_WT_ and ΔNspSEM_WT_ within the protease-substrate complex. (**b**) Distance dynamics between R_17_–C_251_ (substrate) and S_178_–OG_6543_ (protease). (**c**,**d**) Salt bridge formation between substrate R_17_ and protease D_212_/E_214_. (**e**) Proposed “molecular guillotine” mechanism for His_6×_-tag proteolytic cleavage. (**f**) Temporal evolution of substrate-protease hydrogen bond networks.

**Table 1 biomolecules-15-01357-t001:** Purification steps for His_6×_-TCS-ΔNspSEM_WT_ fusion protein expressed in *E. coli*.

Purification Procedures	Volume (mL)	Protein Concentration (mg/mL)	Total Protein (mg)	Total Activity (U)	Specific Activity (U/mg)	Purification Factor (fold)	Yield (%)
Step 1: crude extract from *E. coli* cytosol	25.085	5.32	133.45	6.812	0.051	1	100
Step 2: electrophoretically pure recombinant protein from IMAC	0.800	1.86	1.49	0.158	0.106	2.078	1.12
Step 3: chromatographically pure recombinant protein from SE-HPLC	1.120	0.11	0.12	0.043	0.360	7.059	0.09

## Data Availability

The original contributions presented in this study are included in the article/Appendix A. Further inquiries can be directed to the corresponding authors.

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
