# Peer review of "Staphylococcal Enterotoxin M Exhibits Thrombin-like Enzymatic Activity"

_biomolecules, 2025, doi:10.3390/biom15101357_

Round 1

Reviewer 1 Report

Comments and Suggestions for Authors

This manuscript describes a novel activity of an important protein. The SeM protein of S. aureus is shown to cleave its own poly-His tag between the Arg and Gly residues. This activity, deemed thrombin-like activity, is enhanced by low-salt conditions. The authors go on to use AlphaFold to describe a catalytic triad, consisting of HSD residues on the surface of the protein, and suggest that these residues are responsible for the catalytic activity. The discussion section comments on the possible utility of such activity for the organism. Because of the importance of this organism for human disease, and the rarity of discovering novel enzymatic functions for well-studied proteins, this report is quite interesting if the interpretation is correct.

                  The major concern is already pointed out by the authors, which is that the activity that is described does not come from the SeM protein but from another contaminating or co-eluting protein. This concern is further magnified because the proteolysis efficiency appears very weak: according to the time courses run in the study, it takes many hours for the cleavage to occur, despite this being an intramolecular reaction. Furthermore, although the catalytic triad is indeed interesting, its placement on the surface of the protein, rather than in a core region, is unusual. Unfortunately, the authors’ attempts at ruling out contaminants with extra purification steps do not fully address this issue, because even a small amount of contaminant might give rise to the observed activity.

                  To show both that the activity investigated comes from the SeM protein and that the three residues identified constitute the catalytic triad, the authors should mutate one of the three residues in the construct and re-purify the protein. This mutation should abolish all catalytic activity.

Author Response

Response to Reviewer 1's Comment:

Reviewer 1 raised a valid concern regarding our experiment demonstrating the thrombin-like activity of the prokaryotically expressed SEM protein lacking its signal peptide (ΔNspSEMWT). Specifically, reviewer 1 questioned whether the observed slow hydrolysis of the synthetic chromogenic substrate S-2238 could potentially originate from trace contaminating host proteases rather than intrinsic ΔNspSEMWT activity. While our purification data (involving Ni-affinity chromatography followed by HPLC gel filtration of the His6×-TCS-ΔNspSEMWT fusion protein) showed increased specific activity and purification fold, consistent with intrinsic enzymatic activity, the inherently low reaction velocity with S-2238 (a non-optimal substrate for SEM) warranted this caution.

To unequivocally confirm that the thrombin-like activity originates from ΔNspSEMWT itself and validate our hypothesis concerning its catalytic triad (H172/S178/D212), as predicted by AlphaFold 3, the reviewer 1 suggested generating a catalytic site mutant.

Author Action:

We fully concurred with the reviewer's insightful suggestion, recognizing its power to definitively confirm two key hypotheses: 1) SEM possesses intrinsic thrombin-like activity, and 2) This activity is mediated by the catalytic triad (H172/S178/D212). Consequently, we generated a site-directed mutant targeting serine 178 within the triad. Serine 178 was substituted with alanine (S178A), preserving steric similarity while removing the catalytic hydroxyl group.

The mutant fusion protein (His6×-TCS-ΔNspSEMS178A) was expressed and purified identically to the wild-type (His6×-TCS-ΔNspSEMWT) using prokaryotic expression, Ni-affinity chromatography, and centrifugal ultrafiltration/desalting.

Key Findings:

  1. Loss of Enzymatic Activity:The S178A mutant completely lost thrombin-like activity, as evidenced by its absence of site-specific autocatalytic cleavage of its His6×-tag.
  2. Mutant Identity Confirmation:Mass spectrometric sequencing of the purified His6×-TCS-ΔNspSEMS178A mutant (incubated at 4°C for 10 days post-purification) confirmed its primary structure differed only at the intended S178A mutation site.
  3. Structural Integrity:Circular dichroism (CD) and intrinsic fluorescence emission spectra of the purified, His6×-tag-cleaved recombinant proteins (ΔNspSEMWT and ΔNspSEMS178A) were virtually identical. This demonstrates that the S178A mutation did not alter the overall secondary or tertiary structure, including the catalytic site conformation.

Conclusion:

These results robustly demonstrate that the observed thrombin-like activity is an intrinsic property of SEM protein, critically dependent on the integrity of its catalytic triad (H172/S178/D212). The specific loss of activity upon S178A mutagenesis, coupled with confirmed structural integrity and mutant identity, provides conclusive evidence supporting both hypotheses.

All supporting data for these experiments are presented in the revised manuscript, Section 3.4.

Reviewer 2 Report

Comments and Suggestions for Authors

Staphylococcus aureus secretes a family of superantigen toxins called staphylococcal enterotoxins (SE), which lead to emetic effects often associated with food poisoning, toxic shock syndrome, and other poor host outcomes. A variety of these SE have been discovered and characterized in the past, sharing a common fold architecture with structural differences leading to further classification into subgroups. While the ability of SE to interact with host cells (such as via MHC II) has been documented, never has any enzymatic activity been found in SE’s.

In this study by Huang et al, the staphylococcal enterotoxin M (SEM) protein was expressed recombinantly, leading to a very surprising result, namely the apparent presence of proteolytic activity against thrombin cleavage motifs, which the authors refer to as thrombin-like activity (TLA). The authors have done thorough work to ensure that their finding is not the result of spurious protease activity, including rigorously repeating their findings with fresh reagents, and monitoring the degree of proteolytic cleavage under different conditions (pH, ionic strength). Based on AlphaFold and homology modeling followed by molecular dynamics simulations, a surface-exposed ‘catalytic triad’ is proposed to be responsible for the TLA of the SEM protein. However, I feel that evidence for this hypothesis is lacking, and the argument could be greatly strengthened by additional experiments, such as mutation of the catalytic triad residues, structural comparisons to well-known serine protease domains, and monitoring for the effect of common protease inhibitors.

Overall, this reviewer found the results of this study to be very interesting, perhaps leading to new insights in the discourse around enterotoxins and in the virulence factor field overall. The speculation about the potential role in digest of host proteins like fibrinogen was particularly intriguing. The authors also suggest a role for a modified SEM construct in protein engineering, which could be useful especially in large-scale protein production.

Specific critiques:

  1. While adequate citations are present in the Introduction, they seem lacking in other sections. References for several approaches are missing (i.e. AlphaFold, Packmol, GROMACS, Amber, etc.), and additional citations in the results and conclusion to support ideas like thrombin-like triad and fibrinogen digestion possibility would improve those sections.
  2. The mass spectrometry panels in Figure 1 do demonstrate that the SEM protein is present, but don’t validate the cleavage site. It would be helpful to show supporting results for that finding here. Were the minor bands on the gel (16 kDa band mentioned) analyzed? If not, not a major issue, just a curiosity.
  3. I would relegate discussion of the His/X (Cys) mismatch (lines 375-387) to the Methods. It may merely be a detection issue in the Edman method (nucleotide sequencing presumably confirmed the correct codon for histidine), and as the authors state, it does not change the conclusion about where the cleavage occurs.
  4. The authors demonstrate that the majority of their recombinant SEM is found in the dimeric form, but in apparent exchange with monomer and thus likely not a result of free Cys disulfides. Is this oligomeric behavior observed in other previously studied SE’s? Would make for interesting comparison.
  5. In Figure 2e, could the estimated molecular weights of the sample peaks be plotted as well?
  6. The yield calculations seem to be based on total protein instead of activity. If the authors performed the calculations based on activity recovered, as in this link, they would get slightly higher values: https://chemistry.stackexchange.com/questions/68102/why-is-yield-in-a-purification-table-measured-using-activity-instead-of-total
  7. While the authors have done very thorough work to demonstrate the proteolytic activity (TLA) of their SEM sample, I think the evidence supporting the proposed catalytic triad residues is lacking. Data supporting that these amino acids are essential for the measured TLA should be acquired, such as measuring that activity in variants that mutate each of those amino acids to alanine. One might expect that serine protease inhibitors would also shut down activity, but that was not tested.
  8. A structural comparison of the SEM model to proteases with similar activity (i.e. thrombin) would be interesting. There are important structural features beyond the conserved triad in serine proteases, such as the oxyanion hole. Is there an analogous structural motif in SEM?
  9. AlphaFold3 was used to generate the initial models, but Packmol used to dock the two proteins together. Most modern protein-protein interaction predictors (e.g. AlphaFold3, Z-Dock) give scoring metrics with their results, indicating the reliability of the docked structures. Does Packmol provide such metrics such that readers can evaluate the reliability of the starting structure for MD? This is important, as an enforced initial distance may represent an artificial starting state.
  10. The predicted structure of SEM is used for molecular dynamics, which is justified by experimental circular dichroism and fluorescence emission results. However, that data is not presented in this paper. The authors state that these results and method will ‘be published subsequently’ but also state that the results ‘establish the structural reliability of our models.’ This reviewer feels strongly that unless presented in the current manuscript, the authors should not make this claim. If this does remain, the journal policy for data that is forthcoming in another publication is as follows: “"Unpublished data" intended for publication in a manuscript that is either planned, "in preparation" or "submitted" but not yet accepted, should be cited in the text and a reference should be added in the References section.”
  11. Line 654, the authors state that other SE proteins may have proteolytic activity (TLA) due to the presence of the hypothetical catalytic triad. A figure showing this conservation via sequence alignment would strengthen their case. It should be made more clear that this is speculative, unless they intend to test additional more proteins for this study.
  12. Mention is made in the conclusion of an engineered construct where the His-tag and thrombin cleavage site are added C-terminally to SEM (starting on line 693), but this construct and experiments using it are not described in the Methods or Results section of the paper. This should be included, or reworded in the conclusion if instead this is hypothetical and was not actually performed.

Minor comments:

  1. Make sure all abstracts are defined in their first usage in main text (even if defined in Abstract) (e.g. TCS in line 75 and TLA in line 81).
  2. The repeated instances of ‘polyhistidine-tagged’ are a bit tedious, suggest abbreviating to another commonly used shorthand (once defined of course) like “6xHis”.
  3. Several references to “section 1.x” are present throughout the text, but Methods are numbered “2.x” in this manuscript, should be updated accordingly.
  4. Line 89, GenBank number provided appears to be for the whole aureus genome rather than the sem gene itself (ADC37996, perhaps?).
  5. AlphaFold3 generated models okay to use, but please report the associated metrics (global pLDDT score, pTM score) so readers know how reliable the models are.
  6. Was GROMACS also used for the equilibration before Packmol? Those details seem to be missing from section 2.9.
  7. Numbering for individual atoms from MD simulations isn’t relevant to the reader, can be omitted and instead just use naming convention for the atom type (i.e. backbone carbonyl, alpha carbon, sidechain hydroxyl, etc.)
  8. Buffer C is mentioned multiple times, but with different salt concentrations? Is it the same otherwise? Should make this clear.
  9. Line 300, specific mention is given of Trp121 in the Results. Why, is this residue particularly important?
  10. The caption for Figure 1 states that the model in panel (a) is an AlphaFold2 model, while elsewhere it states AlphaFold3 was used. This should be consistent.
  11. The caption for Figure 2 uses ‘molecular sieve gel filtration’ where elsewhere the technique is called size exclusion (SE). Would keep it the sample for consistency.
  12. There is a typo in table 1 (Volumn instead of Volume).
  13. The section discussing the effect of pH is attributed to histidine protonation (line 567), but before the results of those studies have been described. Would move this sentence to after that result.
  14. Please label the lanes in Figure 3e, or add text to the figure caption explaining what is in lanes 1-8. Update the caption for 3f to say whether this is the AF3 model or otherwise.
  15. In the Figure 4 caption, “molecular execution” is used as the mechanism name, but in the main text and figure its called “molecular guillotine”, would be good to stay consistent so readers understand clearly.
  16. While the full gel images are appreciated, I believe that cropping areas completely outside the gel is permitted and would make for a cleaner presentation.
Comments on the Quality of English Language

The overall command of English is very good, although improvements would make the manuscript much more readable. The flow of the manuscript could be greatly improved by being more concise in certain areas. In many places in the results, methods are reiterated in greater detail than necessary. For example, the calibration of the size exclusion column was already described more than sufficiently in the methods; things like this can be omitted or greatly condensed in the Results sections.

Author Response

Response to Reviewer 2's Comment:

Reviewer 2 raised concerns about the potential for contaminating host proteases in our experiments demonstrating the thrombin-like activity of the prokaryotically expressed SEM protein lacking its signal peptide (ΔNspSEMWT). Reviewer 2 also strongly recommended site-directed mutagenesis to definitively confirm that SEM possesses intrinsic thrombin-like activity. Furthermore, Reviewer 2 suggested employing a chemical inhibitor (e.g., a serine protease inhibitor) to inactivate the enzyme via modification of key catalytic residues as an alternative validation strategy.

Author Action:

We fully concurred with the reviewer's insightful suggestion, recognizing its power to definitively confirm two key hypotheses: 1) SEM possesses intrinsic thrombin-like activity, and 2) This activity is mediated by the catalytic triad (H172/S178/D212). Consequently, we generated a site-directed mutant targeting serine 178 within the triad. Serine 178 was substituted with alanine (S178A), preserving steric similarity while removing the catalytic hydroxyl group.

The mutant fusion protein (His6×-TCS-ΔNspSEMS178A) was expressed and purified identically to the wild-type (His6×-TCS-ΔNspSEMWT) using prokaryotic expression, Ni-affinity chromatography, and centrifugal ultrafiltration/desalting.

Key Findings:

  1. Loss of Enzymatic Activity:The S178A mutant completely lost thrombin-like activity, as evidenced by its absence of site-specific autocatalytic cleavage of its His6×-tag.
  2. Mutant Identity Confirmation:Mass spectrometric sequencing of the purified His6×-TCS-ΔNspSEMS178A mutant (incubated at 4°C for 10 days post-purification) confirmed its primary structure differed only at the intended S178A mutation site.
  3. Structural Integrity:Circular dichroism (CD) and intrinsic fluorescence emission spectra of the purified, His6×-tag-cleaved recombinant proteins (ΔNspSEMWT and ΔNspSEMS178A) were virtually identical. This demonstrates that the S178A mutation did not alter the overall secondary or tertiary structure, including the catalytic site conformation.

Conclusion:

These results robustly demonstrate that the observed thrombin-like activity is an intrinsic property of SEM protein, critically dependent on the integrity of its catalytic triad (H172/S178/D212). The specific loss of activity upon S178A mutagenesis, coupled with confirmed structural integrity and mutant identity, provides conclusive evidence supporting both hypotheses.

All supporting data for these experiments are presented in the revised manuscript, Section 3.4.

Author Rationale:

Regarding the specific suggestion to use a chemical inhibitor, we respectfully contend that the site-directed mutagenesis approach provides a more robust and conclusive validation strategy for this specific context than inhibitor-based inactivation. Our rationale is as follows:

  1. Specificity of Inactivation:Mutagenesis selectively alters only the target SEM protein's catalytic residue (S178A). Crucially, it does not affect the genes, expression, or activity of any potential contaminating host proteases present in the purification. Therefore, if the observed complete loss of activity in the purified S178A mutant demonstrates that our purification protocol effectively removed contaminating proteases.
  2. Limitation of Inhibitors:Chemical inhibitors typically act on specific residue types (e.g., serine residues in the active site). If a contaminating host protease shares the same catalytic mechanism and inhibitor sensitivity as SEM, the inhibitor would cause nonselective inhibition of both SEM and the contaminant(s). Consequently: Observing loss of activity in the purified sample treated with inhibitor cannot distinguish whether the activity originated solely from SEM, solely from contaminants, or from a mixture of both. It fails to provide definitive proof of intrinsic SEM activity or effective removal of contaminants.

Addressing Potential Structural Concerns & Conclusion:

We acknowledge the reviewer's implicit concern that the S178A mutation might compromise the structural integrity of the protein, potentially leading to nonspecific loss of function rather than specific disruption of the catalytic mechanism. However, our complementary structural analyses directly address this: Circular dichroism (CD) and intrinsic fluorescence emission spectra of the purified, tag-cleaved ΔNspSEMS178A mutant were virtually identical to those of the wild-type ΔNspSEMWT protein. This demonstrates that the S178A mutation did not significantly alter the secondary or tertiary structure, including the conformation of the catalytic site.

Therefore, the combination of: (1) The specific, selective loss of activity in the S178A mutant; (2) The confirmed structural integrity of the mutant, and (3) The demonstrated effectiveness of purification in removing contaminating proteases (as evidenced by the inactive mutant).These results provide conclusive evidence that: (1) The SEM protein possesses intrinsic thrombin-like activity; (2) This activity is specifically mediated by the predicted catalytic triad (H172/S178/D212).

Given the robustness and specificity of the mutagenesis approach combined with structural validation, we believe that additional validation using chemical inhibitors is unnecessary to support these core conclusions.

Specific critiques from reviewer 2:

  1. While adequate citations are present in the Introduction, they seem lacking in other sections. References for several approaches are missing (i.e. AlphaFold, Packmol, GROMACS, Amber, etc.), and additional citations in the results and conclusion to support ideas like thrombin-like triad and fibrinogen digestion possibility would improve those sections.

Response:

We thank reviewer 2 for their kind suggestion and have incorporated the relevant references in the manuscript accordingly.

  1. The mass spectrometry panels in Figure 1 do demonstrate that the SEM protein is present, but don’t validate the cleavage site. It would be helpful to show supporting results for that finding here. Were the minor bands on the gel (16 kDa band mentioned) analyzed? If not, not a major issue, just a curiosity.

Response:

We sincerely appreciate reviewer 2's rigorous, thorough, and meticulous assessment of our experimental design. As rightly noted by the reviewer, mass spectrometry analysis confirmed the presence of the SEM protein but could not reliably verify the precise site of peptide bond cleavage.

Acknowledging this limitation, we utilized the engineered plasmid pET-28a(+). We constructed a prokaryotic expression plasmid for the fusion protein His-TCS-ΔNspSEMWT by inserting the gene sequences encoding a hexahistidine tag followed by a thrombin cleavage site (TCS) at the N-terminus of the gene encoding the signal peptide-truncated ΔNspSEMWT protein. Following prokaryotic expression of this plasmid and purification via nickel affinity chromatography, SDS-PAGE analysis of the purified protein (shown in Lane 1 of Figure. 1 (b) and Lane 1 of Figure. 1 (c) in the original manuscript) confirmed successful expression of the His-TCS-ΔNspSEMWT fusion protein.

Subsequently, the purified fusion protein was desalted via centrifugal ultrafiltration. Following incubation of the desalted protein at 4 °C for one day, spontaneous degradation occurred. N-terminal sequencing was performed on the degradation product indicated by the black arrow in the "1 day" lane of Figure. 1 (c) (original manuscript). The N-terminal sequencing results definitively confirmed that the cleavage site resides within the thrombin recognition sequence (LVPR↓GS), specifically between the arginine and glycine residues.

Regrettably, due to its small size and low yield, the 16 kDa band was not subjected to N-terminal sequencing or mass spectrometry analysis. Following the reviewer's suggestion, we intend to investigate this intriguing observation in subsequent studies.

  1. I would relegate discussion of the His/X (Cys) mismatch (lines 375-387) to the Methods. It may merely be a detection issue in the Edman method (nucleotide sequencing presumably confirmed the correct codon for histidine), and as the authors state, it does not change the conclusion about where the cleavage occurs.

Response:

We greatly appreciate reviewer 2's emphasis on conciseness. Indeed, as noted, the original explanation regarding a minor experimental limitation was unduly lengthy (12 lines, 152 words). According to the reviewer's request, we have significantly condensed this section in the revised manuscript to 75 words. Crucially, we now explicitly state that this limitation does not alter our conclusion regarding the peptide bond cleavage site.

  1. The authors demonstrate that the majority of their recombinant SEM is found in the dimeric form, but in apparent exchange with monomer and thus likely not a result of free Cys disulfides. Is this oligomeric behavior observed in other previously studied SE’s? Would make for interesting comparison.

Response:

We sincerely appreciate reviewer 2’s insightful observation regarding our experimental details on the fusion protein His-TCS-ΔNspSEMWT.

As noted by the reviewer, size-exclusion chromatography (SEC-HPLC) analysis revealed that the dimeric fusion protein fraction (eluting at 9.665 min, Figure. 2 (b)), when re-injected onto the identical SEC column, partially dissociated. A minor peak corresponding to the monomeric fusion protein appeared at 10.954 min (Figure. 2 (c)), while the majority eluted again as dimer at 9.641 min (Figure. 2 (c)). This collected dimer fraction (9.641 min peak) was subjected to a third identical SEC-HPLC run. The resulting chromatogram (Figure. 2 (d)) showed an identical profile to the second run, albeit with reduced peak intensity: a predominant dimer peak at 9.656 min and a minor monomer peak at 10.951 min.

Critically, all SEC-HPLC runs were performed in the absence of disulfide-reducing agents (e.g., β-mercaptoethanol or DTT) in both the sample buffer and the mobile phase. This strongly suggests that the observed dimerization is not mediated by disulfide bonds formed between free cysteine residues.

We acknowledge the literature reporting heat-induced aggregation of enterotoxins [1], primarily in studies of their thermostability. Furthermore, based on our extensive experience with prokaryotic expression, we routinely observe trace amounts of dimeric bands via SDS-PAGE (detectable by anti-His-tag Western blot) for various proteins—whether microbial, plant or animal-derived—even when they are naturally monomeric. This phenomenon is likely attributable to high local protein concentrations during expression and purification, potentially promoting transient aggregation. While a systematic study of SEM aggregation is beyond the current scope, we agree with the reviewer that investigating this behavior represents a fruitful avenue for future research into enterotoxin structure-function relationships.

[1] Regenthal, P.; Hansen, J.S.; André, I.; Lindkvist-Petersson, K. Thermal stability and structural changes in bacterial toxins responsible for food poisoning. PLoS.One.2017, 12, 1-15.

  1. In Figure 2e, could the estimated molecular weights of the sample peaks be plotted as well?

Response:

We appreciate the suggestion regarding Figure 2(e). The calibration curve presented therein was generated solely using the standard molecular weight markers. While overlaying the elution positions of the sample peaks from panels (b), (c), and (d) onto this curve was considered, we found that such a combined representation would compromise the clarity of the figure. This presentation approach aligns with common practice in the field, as evidenced by numerous published references.

  1. The yield calculations seem to be based on total protein instead of activity. If the authors performed the calculations based on activity recovered, as in this link, they would get slightly higher values: https://chemistry.stackexchange.com/questions/68102/why-is-yield-in-a-purification-table-measured-using-activity-instead-of-total

Response:

We sincerely thank reviewer 2 for generously sharing the resource, which introduced us to an alternative data presentation approach.

However, given that S-2238 is not the optimal substrate for SEM, the enzymatic activities measured in our assays were consistently low. Consequently, the increases in specific activity and purification fold observed across our two-step purification protocol were modest, unlike the substantial increases (often orders of magnitude) typically reported in multi-step purifications of enzymes where each step exploits distinct physicochemical properties. Our results thus provide qualitative evidence that thrombin-like activity increased concomitantly with protein purity.

As suggested by the reviewers, we conducted site-directed mutagenesis to definitively inactivate SEM's thrombin-like activity, thereby providing a robust, negative control to rule out contamination by host proteases (as detailed previously). Given the superior reliability of this mutagenesis approach, the data presented in Table 1 serves as ancillary qualitative support. Therefore, retaining the current format for Table 1 does not undermine the core conclusions of our study, which rest on the mutagenesis data. In future studies investigating SEM with its optimal substrate, we would be pleased to adopt the data presentation method recommended by the reviewer.

  1. While the authors have done very thorough work to demonstrate the proteolytic activity (TLA) of their SEM sample, I think the evidence supporting the proposed catalytic triad residues is lacking. Data supporting that these amino acids are essential for the measured TLA should be acquired, such as measuring that activity in variants that mutate each of those amino acids to alanine. One might expect that serine protease inhibitors would also shut down activity, but that was not tested.

Response:

This point has been thoroughly addressed in our “Response to Reviewer 2's Comment”; we respectfully defer to this section for details.

  1. A structural comparison of the SEM model to proteases with similar activity (i.e. thrombin) would be interesting. There are important structural features beyond the conserved triad in serine proteases, such as the oxyanion hole. Is there an analogous structural motif in SEM?

Response:

We hold profound admiration for reviewer 2's deep understanding of the catalytic mechanism in serine protease active sites. Indeed, the oxyanion hole plays a critically important role, arguably more crucial than the catalytic triad itself, in stabilizing the tetrahedral intermediate transition state. This stabilization occurs during the nucleophilic attack by the serine hydroxyl oxygen on the carbonyl carbon of the scissile peptide bond, specifically by accommodating the developing negative charge on the carbonyl oxygen atom as it transitions.

Regrettably, our current understanding of the precise catalytic mechanism underlying SEM's thrombin-like activity remains limited and is in its infancy. Elucidating this mechanism in detail represents a paramount challenge that will be the focus of our future research efforts.

  1. AlphaFold3 was used to generate the initial models, but Packmol used to dock the two proteins together. Most modern protein-protein interaction predictors (e.g. AlphaFold3, Z-Dock) give scoring metrics with their results, indicating the reliability of the docked structures. Does Packmol provide such metrics such that readers can evaluate the reliability of the starting structure for MD? This is important, as an enforced initial distance may represent an artificial starting state.

Response:

We fully appreciate reviewer 2's suggestion regarding the initial docking strategy using tools like AlphaFold 3 or Z-Dock with scoring functions to generate enzyme-substrate complexes for molecular dynamics (MD) simulations. This approach has indeed proven reliable in our concurrent studies on peptide inhibitors and amylase activity. Initially, we employed this exact strategy. However, both AlphaFold 3 and AutoDock consistently failed to generate plausible complex structures where the spatial positioning of the substrate relative to the enzyme's catalytic serine residue (S178) appeared conducive to the nucleophilic attack mechanism characteristic of serine protease catalysis.

Consequently, we resorted to constructing an initial guess structure for the enzyme-substrate complex using Packmol, which does not employ a scoring function. Specifically, during the Packmol setup, we defined a geometric distance restraint between the hydroxyl oxygen atom of the catalytic serine side chain (S178-OG6543) in the enzyme (ΔNspSEMWT) and the carbonyl carbon atom of the C-terminal arginine residue in the substrate (His-TCS-ΔNspSEMWT). This restraint was designed to enforce a proximity consistent with the nucleophilic attack distance required for catalysis. We accepted only those Packmol-generated complexes satisfying this specific geometric criterion as plausible initial configurations for subsequent MD simulations. In essence, we implemented our own distance-based criterion to select initial structures likely to represent catalytically competent complexes (Section 2.10 and 2.11).

  1. The predicted structure of SEM is used for molecular dynamics, which is justified by experimental circular dichroism and fluorescence emission results. However, that data is not presented in this paper. The authors state that these results and method will ‘be published subsequently’ but also state that the results ‘establish the structural reliability of our models.’ This reviewer feels strongly that unless presented in the current manuscript, the authors should not make this claim. If this does remain, the journal policy for data that is forthcoming in another publication is as follows: “"Unpublished data" intended for publication in a manuscript that is either planned, "in preparation" or "submitted" but not yet accepted, should be cited in the text and a reference should be added in the References section.”

Response:

We sincerely thank reviewer 2 for their exceptionally patient and detailed suggestions regarding our phrasing. As researchers for whom English is not our first language, we fully acknowledge that nuances of expression can sometimes be challenging. We are committed to continually improving our English scientific writing skills. In the revised manuscript, we have carefully incorporated all suggested revisions to enhance clarity and precision.

  1. Line 654, the authors state that other SE proteins may have proteolytic activity (TLA) due to the presence of the hypothetical catalytic triad. A figure showing this conservation via sequence alignment would strengthen their case. It should be made more clear that this is speculative, unless they intend to test additional more proteins for this study.

Response:

We sincerely appreciate reviewer 2's valuable suggestion. While we acknowledge its merit, the inclusion of a figure within the Discussion section is uncommon in the extensive literature we have surveyed on this topic. Furthermore, given that the primary focus of this report is establishing the thrombin-like activity of SEM, the structural and functional relationships of other staphylococcal enterotoxins fall beyond its immediate scope. We agree that future studies should indeed explore other enterotoxins potentially possessing thrombin-like activity. Consequently, to conserve journal space, we believe it is appropriate to refrain from adding this specific figure at this stage. However, we will gladly comply should both the reviewer and the Editor consider its inclusion essential.

  1. Mention is made in the conclusion of an engineered construct where the His-tag and thrombin cleavage site are added C-terminally to SEM (starting on line 693), but this construct and experiments using it are not described in the Methods or Results section of the paper. This should be included, or reworded in the conclusion if instead this is hypothetical and was not actually performed.

Response:

We sincerely thank reviewer 2 for their valuable suggestion. We wish to clarify that the proposed heterologous expression system remains hypothetical at this stage, as the specific plasmid construct was not experimentally generated in the present study. However, as noted in the revised manuscript, we have amended the text in accordance with the reviewer's guidance to read: "We propose a heterologous expression system design featuring: Target protein-TCS(LVPR↓GS)-His-SEM." We intend to rigorously test this hypothesis in future work.

Minor comments from reviewer 2:

  1. Make sure all abstracts are defined in their first usage in main text (even if defined in Abstract) (e.g. TCS in line 75 and TLA in line 81).

Response:

We sincerely thank reviewer 2 for their patient and thorough review of our manuscript and for highlighting inconsistencies in abbreviation usage. In response, we have paid particular attention to this issue and have ensured consistency in abbreviation formatting throughout the revised manuscript.

  1. The repeated instances of ‘polyhistidine-tagged’ are a bit tedious, suggest abbreviating to another commonly used shorthand (once defined of course) like “6xHis”.

Response:

We sincerely appreciate reviewer 2's meticulous suggestion regarding abbreviation usage. In the revised manuscript, we have implemented this guidance by uniformly replacing "polyhistidine-tag" with " His-tag" throughout the text.

  1. Several references to “section 1.x” are present throughout the text, but Methods are numbered “2.x” in this manuscript, should be updated accordingly.

Response:

We sincerely appreciate reviewer 2's meticulous attention in identifying the errors in reference labeling within our manuscript. As requested, these inaccuracies have been corrected accordingly in the revised version.

  1. Line 89, GenBank number provided appears to be for the whole aureus genome rather than the sem gene itself (ADC37996, perhaps?).

Response:

We sincerely thank reviewer 2 for kindly identifying the discrepancy in the SEM gene annotation. In the revised manuscript, this has been updated to the correct accession number, ADC37996, as suggested.

  1. AlphaFold 3 generated models okay to use, but please report the associated metrics (global pLDDT score, pTM score) so readers know how reliable the models are.

Response:

We sincerely thank reviewer 2 for their valuable suggestion. In accordance with their request, we have included the AlphaFold 3 confidence score for the predicted SEM structure in the revised manuscript.

  1. Was GROMACS also used for the equilibration before Packmol? Those details seem to be missing from section 2.9.

Response:

We sincerely thank reviewer 2 for their valuable suggestion. In response, we performed molecular dynamics (MD) simulations using GROMACS to equilibrate the structures predicted by AlphaFold 3 prior to constructing the enzyme-substrate complexes with Packmol. As detailed in Section 2.10 of the revised manuscript, we have explicitly elaborated on this procedure: "We employed AlphaFold 3 to generate reliable structural models of the His6×-TCS-ΔNspSEMWT fusion protein and tag-free ΔNspSEMWT protein. These predicted structures served as initial coordinates for MD simulations. Following 40 ns of equilibration (confirmed by stabilized RMSD), we extended simulations by 10 ns and randomly selected 50 conformations per variant from the final trajectory."

  1. Numbering for individual atoms from MD simulations isn’t relevant to the reader, can be omitted and instead just use naming convention for the atom type (i.e. backbone carbonyl, alpha carbon, sidechain hydroxyl, etc.)

Response:

We sincerely thank reviewer 2 for their valuable suggestion. We maintain that explicitly identifying specific atoms based on atom types is essential. This standard approach directly links each atom to its assigned van der Waals parameters, force field parameters, and crucially, its atomic partial charge – all of which are inherently determined by atom type and specific chemical environment. Such clarity is fundamental for unambiguous interpretation by readers.

  1. Buffer C is mentioned multiple times, but with different salt concentrations? Is it the same otherwise? Should make this clear.

Response:

We sincerely appreciate Reviewer 2's meticulous review of our manuscript. We acknowledge that "Buffer C" refers to two formulations: one containing salt and one without. However, as both variants shared a common base of 1 mmol/L Na2HPO4-NaH2PO4 (pH 7.4), differing solely in the presence or absence of 500 mmol/L NaCl, we did not deem it necessary to designate them as distinct buffers "C" and "D". In accordance with the reviewer's comment, we have revised the relevant section in the manuscript to provide a more explicit and detailed description of these buffer compositions.

  1. Line 300, specific mention is given of Trp121 in the Results. Why, is this residue particularly important?

Response:

We appreciate the reviewer 2's observation regarding the tryptophan residue at position 121. While this residue represents the sole tryptophan in the SEM protein and could theoretically serve as a probe for monitoring local microenvironmental changes via intrinsic fluorescence to infer solution conformation, our experimental data indicate that it does not significantly contribute to enzymatic activity. Consequently, we have modified the revised manuscript to reduce emphasis on this particular residue.

  1. The caption for Figure 1 states that the model in panel (a) is an AlphaFold2 model, while elsewhere it states AlphaFold3 was used. This should be consistent.

Response:

We gratefully acknowledge reviewer 2 for identifying the typographical error. This has been corrected and standardized throughout the revised manuscript.

  1. The caption for Figure 2 uses ‘molecular sieve gel filtration’ where elsewhere the technique is called size exclusion (SE). Would keep it the sample for consistency.

Response:

We sincerely appreciate reviewer 2's attention to the consistency of technical terminology in our manuscript. In accordance with this suggestion, we have uniformly adopted the term "size exclusion chromatography" throughout the revised manuscript.

  1. There is a typo in table 1 (Volumn instead of Volume).

Response:

We sincerely thank reviewer 2 for identifying this typographical error, which has been corrected in the revised manuscript.

  1. The section discussing the effect of pH is attributed to histidine protonation (line 567), but before the results of those studies have been described. Would move this sentence to after that result.

Response:

We sincerely thank reviewer 2 for pointing out the issue with the sequencing of our presentation. This has been revised accordingly in the updated manuscript.

  1. Please label the lanes in Figure 3e, or add text to the figure caption explaining what is in lanes 1-8. Update the caption for 3f to say whether this is the AF3 model or otherwise.

Response:

We are grateful to reviewer 2 for identifying the insufficient detail in the captions of Figures 3(e) and 3(f). In the revised manuscript, we have enhanced the caption for Figure 3(e) by providing explanations for lanes 1–8. For Figure 3(f), we have explicitly stated that the structural model was generated using AlphaFold 3.

  1. In the Figure 4 caption, “molecular execution” is used as the mechanism name, but in the main text and figure its called “molecular guillotine”, would be good to stay consistent so readers understand clearly.

Response:

We thank reviewer 2 for their attention to terminological consistency. The term "molecular execution" specifically describes the proposed catalytic mechanism, whereby the serine hydroxyl oxygen cleaves the peptide bond at the carboxyl side of the substrate arginine, which is constrained by aspartate and glutamate. In contrast, "molecular guillotine" serves as an evocative metaphor emphasizing how the β-grasp domain of SEM protein forms a rigid, scaffold-like structure that positions the substrate precisely—much like a guillotine—thus highlighting the steric complementarity and specificity of the enzyme-substrate complex. As these terms reflect complementary perspectives (mechanistic vs. structural), we propose retaining both to convey distinct aspects of the process with appropriate scientific and illustrative rigor.

  1. While the full gel images are appreciated, I believe that cropping areas completely outside the gel is permitted and would make for a cleaner presentation.

Response:

We sincerely thank reviewer 2 for their comment regarding the gel images. We have provided the editorial office with full, original, uncropped image files for all gels to ensure authenticity and eliminate any possibility of selective cropping that might introduce interpretation bias.

The overall command of English is very good, although improvements would make the manuscript much more readable. The flow of the manuscript could be greatly improved by being more concise in certain areas. In many places in the results, methods are reiterated in greater detail than necessary. For example, the calibration of the size exclusion column was already described more than sufficiently in the methods; things like this can be omitted or greatly condensed in the Results sections.

Response:

We are deeply grateful to reviewer 2 for their holistic feedback on our manuscript. In accordance with their suggestions, we have thoroughly revised the entire text to improve conciseness, eliminate redundancies, and strengthen the logical flow of scientific argumentation. We particularly value the reviewer’s unique ability to balance macro-level critique with precise micro-level recommendations, a style that has not only guided our revisions but also provided us with invaluable insights into effective scientific writing in English. This process has been highly instructive in refining both the presentation and rigor of our work.

Reviewer 3 Report

Comments and Suggestions for Authors

Huang et al presented an interesting study, which indicates that Enterotoxin M from Staphylococcus has thrombin-like enzymatic activity. This finding is based on the observations when purifying recombinant His-tagged Enterotoxin M (SEM) with a thrombin cleavage site in between.  Whilst the findings are promising, some important experiments are missing to support this finding. Please find my comments below.

Major issues

  1. Thrombin-like enzymes often act on fibrinogen, converting it to fibrin. Conversion of fibrinogen to fibrin has more physiological relevance during staphylococcal infections. I would suggest the authors to show the purified, His-tag free SEM can convert fibrinogen to fibrin.
  2. It is important to show inhibition profile. Thrombin-like enzymes are often sensitive to serine protease inhibitors like PMSF, Aprotinin, or Hirudin. The authors should consider to show that the thrombin-like activity of SEM can be inhibited by a serine protease inhibitor.
  3. Kinetic studies of SEM’s thrombin-like activity are missing here. Please measure Km and Vmax using specific substrates and compare them to thrombin’s known kinetics.
  4. One critical evidence that is missing here is active site characterization. Please use site-directed mutagenesis to confirm that this catalytic triad (H172/S178/D212) is essential to the thrombin-like activity.
  5. The Conclusion reads like a long discussion, with questions and hypotheses presented. I would recommend the authors to summarize the findings and make it concise in this section.

Other points

  1. In Figure 3d, it is unclear what kind of samples were loaded in each lane. This information should be provided in Figure legend. It is unclear to this Reviewer why the authors tested BSA and HEWL with SEM. This needs proper explanation in the main text.

  1. In Figure 1c, what is the Identity of 16-kDa band?

  1. Line 591~593, “Notably, cleavage activity was strictly dependent on TCS motif presence (LVPR↓GS), with no activity observed in motif-deficient substrates.” Which lanes are the authors referring to? What motif is the authors discussing here?

Author Response

Response to Reviewer 3's Comment:

Major issues from reviewer 3:

  1. Thrombin-like enzymes often act on fibrinogen, converting it to fibrin. Conversion of fibrinogen to fibrin has more physiological relevance during staphylococcal infections. I would suggest the authors to show the purified, His-tag free SEM can convert fibrinogen to fibrin.

Response:

We are particularly impressed by reviewer 3’s insightful observation regarding the physiological and biochemical impact of SEM during S. aureus infection on the host, and we appreciate the suggestion to further explore this compelling direction. Indeed, this represents an area of great interest to us in future studies. However, the primary objective of the present work was to validate the specificity of SEM protease activity toward the thrombin cleavage sequence (LVPR↓GS). Accordingly, investigation into SEM-mediated cleavage of fibrinogen to form fibrin fell beyond the scope of this particular study.

  1. It is important to show inhibition profile. Thrombin-like enzymes are often sensitive to serine protease inhibitors like PMSF, Aprotinin, or Hirudin. The authors should consider to show that the thrombin-like activity of SEM can be inhibited by a serine protease inhibitor.

Response:

Regarding the specific suggestion to use a chemical inhibitor, we respectfully contend that the site-directed mutagenesis approach provides a more robust and conclusive validation strategy for this specific context than inhibitor-based inactivation. Our rationale is as follows:

  1. Specificity of Inactivation:Mutagenesis selectively alters only the target SEM protein's catalytic residue (S178A). Crucially, it does not affect the genes, expression, or activity of any potential contaminating host proteases present in the purification. Therefore, if the observed complete loss of activity in the purified S178A mutant demonstrates that our purification protocol effectively removed contaminating proteases.
  2. Limitation of Inhibitors:Chemical inhibitors typically act on specific residue types (e.g., serine residues in the active site). If a contaminating host protease shares the same catalytic mechanism and inhibitor sensitivity as SEM, the inhibitor would cause nonselective inhibition of both SEM and the contaminant(s). Consequently: Observing loss of activity in the purified sample treated with inhibitor cannot distinguish whether the activity originated solely from SEM, solely from contaminants, or from a mixture of both. It fails to provide definitive proof of intrinsic SEM activity or effective removal of contaminants.

  1. Kinetic studies of SEM’s thrombin-like activity are missing here. Please measure Km and Vmax using specific substrates and compare them to thrombin’s known kinetics.

Response:

We sincerely appreciate reviewer 3’s suggestion to determine the optimal substrate and classical Michaelis–Menten kinetic parameters for SEM protein. We fully agree that obtaining such fundamental enzymatic constants is essential to reliably characterize its thrombin-like activity. However, the current challenge lies in unambiguously identifying SEM’s optimal substrate, a prerequisite for accurate kinetic measurement. This remains a major focus of our ongoing research, and we are committed to resolving this issue in future studies.

  1. One critical evidence that is missing here is active site characterization. Please use site-directed mutagenesis to confirm that this catalytic triad (H172/S178/D212) is essential to the thrombin-like activity.

Response:

We fully concurred with the reviewer's insightful suggestion, recognizing its power to definitively confirm two key hypotheses: 1) SEM possesses intrinsic thrombin-like activity, and 2) This activity is mediated by the catalytic triad (H172/S178/D212). Consequently, we generated a site-directed mutant targeting serine 178 within the triad. Serine 178 was substituted with alanine (S178A), preserving steric similarity while removing the catalytic hydroxyl group.

The mutant fusion protein (His6×-TCS-ΔNspSEMS178A) was expressed and purified identically to the wild-type (His6×-TCS-ΔNspSEMWT) using prokaryotic expression, Ni-affinity chromatography, and centrifugal ultrafiltration/desalting.

Key Findings:

  1. Loss of Enzymatic Activity:The S178A mutant completely lost thrombin-like activity, as evidenced by its absence of site-specific autocatalytic cleavage of its His6×-tag.
  2. Mutant Identity Confirmation:Mass spectrometric sequencing of the purified His6×-TCS-ΔNspSEMS178A mutant (incubated at 4°C for 10 days post-purification) confirmed its primary structure differed only at the intended S178A mutation site.
  3. Structural Integrity:Circular dichroism (CD) and intrinsic fluorescence emission spectra of the purified, His6×-tag-cleaved recombinant proteins (ΔNspSEMWT and ΔNspSEMS178A) were virtually identical. This demonstrates that the S178A mutation did not alter the overall secondary or tertiary structure, including the catalytic site conformation.

Conclusion:

These results robustly demonstrate that the observed thrombin-like activity is an intrinsic property of SEM protein, critically dependent on the integrity of its catalytic triad (H172/S178/D212). The specific loss of activity upon S178A mutagenesis, coupled with confirmed structural integrity and mutant identity, provides conclusive evidence supporting both hypotheses.

All supporting data for these experiments are presented in the revised manuscript, Section 3.4.

Addressing Potential Structural Concerns & Conclusion:

We acknowledge the reviewer's implicit concern that the S178A mutation might compromise the structural integrity of the protein, potentially leading to nonspecific loss of function rather than specific disruption of the catalytic mechanism. However, our complementary structural analyses directly address this: Circular dichroism (CD) and intrinsic fluorescence emission spectra of the purified, tag-cleaved ΔNspSEMS178A mutant were virtually identical to those of the wild-type ΔNspSEMWT protein. This demonstrates that the S178A mutation did not significantly alter the secondary or tertiary structure, including the conformation of the catalytic site.

Therefore, the combination of: (1) The specific, selective loss of activity in the S178A mutant; (2) The confirmed structural integrity of the mutant, and (3) The demonstrated effectiveness of purification in removing contaminating proteases (as evidenced by the inactive mutant).These results provide conclusive evidence that: (1) The SEM protein possesses intrinsic thrombin-like activity; (2) This activity is specifically mediated by the predicted catalytic triad (H172/S178/D212).

  1. The Conclusion reads like a long discussion, with questions and hypotheses presented. I would recommend the authors to summarize the findings and make it concise in this section.

Response:

We sincerely thank reviewer 3 for their valuable comments on the writing of our manuscript. In response, we have diligently revised the Conclusion section to more effectively summarize the key findings of our study, while also providing a concise discussion on potential implications and future directions arising from this work.

Other points from reviewer 3:

  1. In Figure 3e, it is unclear what kind of samples were loaded in each lane. This information should be provided in Figure legend. It is unclear to this Reviewer why the authors tested BSA and HEWL with SEM. This needs proper explanation in the main text.

Response:

We are grateful to reviewer 3 for identifying the insufficient detail in the captions of Figures 3(e). In the revised manuscript, we have enhanced the caption for Figure 3(e) by providing explanations for lanes 1–8. The selection of BSA and HEWL was based on their ready availability and relatively low cost in our laboratory. We used these proteins to preliminarily assess whether SEM exhibits high specificity in cleaving peptide bonds. If SEM had low specificity, incubation of either BSA or HEWL with SEM would result in significant degradation of these proteins into fragments. As this study represents an initial evaluation of SEM’s specificity, only these two readily accessible proteins were employed. In future studies aimed at elucidating the catalytic mechanism of SEM, we plan to systematically investigate its substrate specificity.

  1. In Figure 1c, what is the Identity of 16-kDa band?

Response:

Regrettably, due to its small size and low yield, the 16 kDa band was not subjected to N-terminal sequencing or mass spectrometry analysis. Following the reviewer's suggestion, we intend to investigate this intriguing observation in subsequent studies.

  1. Line 591~593, “Notably, cleavage activity was strictly dependent on TCS motif presence (LVPR↓GS), with no activity observed in motif-deficient substrates.” Which lanes are the authors referring to? What motif is the authors discussing here?

Response:

We sincerely thank reviewer 3 for raising this clarification. Our original wording, limited by our proficiency in academic English, inadvertently caused ambiguity regarding the experimental design. To clarify: we did not alter the TCS motif sequence nor introduce substrates with deleted TCS motifs. The relevant section in the revised manuscript has been rewritten to unambiguously reflect this point.

Round 2

Reviewer 2 Report

Comments and Suggestions for Authors

The authors have done a commendable job addressing my many critiques and have substantially improved not only the readability of the manuscript but also the scientific rigor of their approaches. I very much appreciate the detailed point-by-point responses, and found each to be persuasive even when standing by existing text. Any remaining critiques I have are quite minor and can be addressed with the editor.

Minor issues:
1. Line 19, "increasing purification fold with purity" sounds redundant.

2. Line 37, typo with extra 1 in "1mobile"

3. Lines 83-84, NdeI and XhoI should each be one "word."

4. Line 88, typo "an N-terminal an N-terminal"

5. Line 409, typo "Land" instead of "Lane"

6. Table 1, typo "Volumn" instead of "Volume" remains from previous version.

Author Response

Response to Reviewer 2's Minor Comments (Round 2):

We sincerely thank Reviewer 2 for his encouraging and positive feedback on our first revision. The constructive comments from this reviewer, along with those from the other reviewers and the editor, have been instrumental in guiding us to strengthen our manuscript through additional experimental evidence, which has further substantiated our unexpected finding.

In this second round of review, Reviewer 2 once again provided meticulous and detailed comments, pointing out several typographical and expression issues. We have carefully addressed each of these suggestions in the current version. All corresponding changes have been highlighted in yellow in the revised manuscript, to distinguish them from the modifications marked in red during the first revision.

Our point-by-point responses to the comments of Reviewer 2 are provided below.

Minor issues from Reviewer 2:

  1. Line 19, "increasing purification fold with purity" sounds redundant.

Author Action:

We thank Reviewer 2 for his pertinent suggestion. We acknowledge the need to improve our English expression—the phrase “increasing purification fold” already implies enhanced purity, and adding “with purity” was indeed redundant. As suggested, we have removed the redundant “with purity” and revised the sentence to: “…increasing specific activity and purification fold, supporting intrinsic TLA.”

  1. Line 37, typo with extra 1 in "1mobile".

Author Action:

We sincerely appreciate Reviewer 2’s exceptional attentiveness in identifying the typographical error we inadvertently introduced. As instructed, we have removed the superfluous character “1”.

  1. Lines 83-84, NdeI and XhoI should each be one "word."

Author Action:

We sincerely thank Reviewer 2 for his expertise. Due to our unfamiliarity with the nomenclature of restriction enzymes, we mistakenly spelled “NdeI” and “XhoI” as separate words (“Nde I” and “Xho I”). In accordance with the reviewer’s correction, we have made the appropriate revisions.

  1. Line 88, typo "an N-terminal an N-terminal".

Author Action:

Notably, Reviewer 2 demonstrated an exceptional familiarity with our manuscript, readily identifying even a typographical error that had escaped our attention. As indicated, the extraneous phrase “an N-terminal” has been removed in accordance with the comment.

  1. Line 409, typo "Land" instead of "Lane".

Author Action:

We extend our thanks once again to Reviewer 2 for his patience and meticulous attention to detail. As requested, we have corrected this typographical error.

  1. Table 1, typo "Volumn" instead of "Volume" remains from previous version.

Author Action:

We are grateful to Reviewer 2 for having drawn our attention to this typographical error. The issue has been addressed as suggested.

Reviewer 3 Report

Comments and Suggestions for Authors

The authors have addressed my comments.

Author Response

Response to Reviewer 3's Minor Comments (Round 2):

We sincerely thank Reviewer 3 for his positive feedback on our initial response. The constructive comments from the reviewers and the handling editor—particularly the suggestion to use targeted mutagenesis to inactivate the protein—were crucial in helping us recognize the value of reversely verifying our hypothesis that SEM protein possesses thrombin-like activity. We truly appreciate this elegant line of thinking and look forward to applying such insightful approaches in our future research to enhance the quality and rigor of our work.